# A Graph Theoretic Approach for Preference Learning with Feature Information

**Aadirupa Saha**[1]                                   **Arun Rajkumar**[2]

[1]Apple, Cupertino, USA
[2]IIT Madras, India

## Abstract

We consider the problem of ranking a set of $n$ items given a sample of their pairwise preferences. It is well known from the classical results of sorting literature that without any further assumption, one requires a sample size of $\Omega(n \log n)$ with active selection of pairs whereas, for a random set pairwise preferences the bound could be as bad as $\Omega(n^2)$. However, what if the learner is exposed to additional knowledge of the items features and their pairwise preferences are known to be modelled in terms of their feature similarities – can these bounds be improved? In particular, we introduce a new probabilistic preference model, called feature-Bradley-Terry-Luce (f-BTL) for the purpose, and present a new least squares based algorithm, fBTL-LS, which requires a sample complexity much lesser than $O(n \log n)$ random pairs to obtain a 'good' ranking. The sample complexity of our proposed algorithms depends on the degree of feature correlation of the items that makes use of tools from classical graph matching theory, shedding light on the true complexity of the problem – this was not possible before with existing matrix completion based tools. We also prove tightness of our results showing a matching information theoretic lower bound for the problem. Our theoretical results are corroborated with extensive experimental evaluations on varying datasets.

## 1 INTRODUCTION

Given a set of $n$ items and $m$ pairwise comparisons among them, the problem of *ranking from pairwise preferences* is to recover an underlying ranking among the $n$ items. This is a well-studied problem in several disciplines including statistics, operations research, theoretical computer science, social choice theory, machine learning, decision systems etc [23, 4, 15, 20, 2], [5, 9, 11, 16], [24, 6, 18, 22, 3, 7, 19, 21,

17]. A typical approach to solve this problem is to assume that the comparisons are generated in a stochastic fashion according to a score based pairwise probability model, e.g. Bradley-Terry-Luce model [4] [15] or the Thurstone model [23] and develop algorithms [9], [16], [18], [3] that first estimate the score vector from the given comparisons and obtain the final ranking by simply sorting their estimated scores.

However in practice they suffer from several shortcomings: Firstly, often times side information such as features or relationships among items are available, e.g. to rank a set of mobile phones, it is natural to use features such as cost, battery life, size etc., which influence the pairwise preferences of users in preferring one mobile over other. However, most algorithms do not take this additional information into account. Secondly, they fail to handle the case when new items get added as one cannot find the position of a new item in an already estimated ranking without collecting at least few pairwise preferences of it. Finally, the sample complexity of previous approaches scale as $O(n \log n)$ which can proved to be sub-optimal when item preferences are based on their feature similarities.

In this work, we introduce the *feature-Bradley–Terry–Luce (f-BTL)* model of pairwise comparisons to tackle the problems listed above. The f-BTL model is a generalization of the standard BTL model where the probability of preferring one item over the other explicitly depends on their associated features such that similar items get similar ranks. We next propose a least squares-based algorithm *fBTL-LS* – the novelty of our approach lies in the sample complexity analysis (i.e. the number of comparisons needed to achieve a fixed error) for recovering a 'near-optimal' ranking. The key ingredient used here is a *relation graph* that we define on the items based on their features correlation and apply ideas from classical graph matching theory on the relation graph. Precisely, our sample complexity bound is of $O(\alpha \log \alpha)$, where on an intuitive level, $\alpha$ denotes the number of the main (independent) items that influence the preference structures of the rest of $n - \alpha$ items in the set—This

*Accepted for the 40th Conference on Uncertainty in Artificial Intelligence*  (UAI 2024).

shows a significant reduction in the number of comparisons needed, compared to the earlier known bound $O(n \log n)$, especially when $\alpha << n$, which often is the case in many applications. Furthermore, we also give a matching sample complexity lower bound analyzing the minimal number of pairwise preferences required, establishing the optimality of our algorithm. Our experimental evaluation shows the proposed algorithm significantly outperforms existing algorithms, demonstrating its usefulness on various special types of relation graphs including union of cliques, disconnected graphs, trees, stars, cycles, etc. Our *contributions* are listed below:

**1.** We introduce a new probabilistic model, f-BTL, for ranking from pairwise comparisons which explicitly uses features associated with items (Sec. 2).

**2.** We give a novel sample complexity analysis using ideas from graph matching theory that captures the dependencies among features explicitly in terms of structural properties of the graph, unlike previous approaches (Sec. 3).

**3.** We propose an algorithm, *fBTL-LS* and provide its sample complexity guarantees for recovering a 'good estimate' of the score vector under f-BTL (Sec. 4).

**4.** We finally show our sample complexity guarantee is tight proving a matching lower bound (Sec. 5).

**5.** Our experimental results support our theoretical findings showing the superiority of our algorithm on both synthetic data and real datasets (Sec. 6).

## 2 PRELIMINARIES AND SETTING

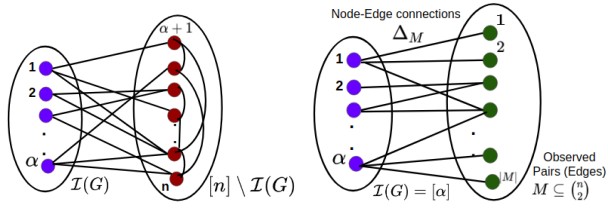

(a) Relation graph $G([n], E)$ associated to $\mathbf{U}$ (Sec. 2.1)

(b) The bipartite graph $C_M = (\mathcal{I}(G) \cup M, \Delta_M)$ (Thm. 3.1)

Figure 1: Few graphical demonstrations

**Notations.** We use lowercase boldface letters for vectors, uppercase boldface letters for matrices, lowercase letters for scalars and uppercase letters for constants. $\| \cdot \|_2$ denotes the $\ell_2$ norm for vectors and spectral norm for matrices. $\| \cdot \|_F$ denotes the Frobenius norm for matrices. We denote the set $\{1, \dots, n\}$ by $[n]$. For any matrix $\mathbf{A} \in \mathbb{R}^{m \times n}$, we abbreviate $A_{ij} = A(i, j)$.

**Bradley-Terry-Luce (BTL) model.** ([4], [15]) A standard probabilistic model for pairwise comparisons is the Bradley-Terry-Luce (BTL) model where the probability of preferring item $i$ over $j$ is given by: $P_{ij} = \frac{\exp(\theta_i)}{\exp(\theta_i) + \exp(\theta_j)}$, $\boldsymbol{\theta} \in \mathbb{R}^n$

being the '*score vector*' of the $n$ items.

### 2.1 PROBLEM SETTING

Let $[n] = \{1, \dots, n\}$ be the set of items to be ranked, and their feature vectors are $\mathbf{U} = \{\mathbf{u}_1, \dots, \mathbf{u}_n\} \subset \mathbb{R}^d$, where $\mathbf{U}$ respects a *relation graph* $G([n], E)$ as follows:

**Feature ($\mathbf{U}$) vs Relation graph ($G$).** We assume the feature set $\mathbf{U}$ of the $n$ items are associated to an underlying relation graph $G([n], E)$ by a natural assumption: $G([n], E)$ is such that there exists an independent set of $G$, say $\mathcal{I}(G)$, such that the set of item features $\mathbf{U} = \{\mathbf{u}_i\}_{i \in [n]}$ lies in the linear span of only that of $\mathcal{I}(G)$, $\{\mathbf{u}_i\}_{i \in \mathcal{I}(G)}$. More formally,

$$\mathbf{u}_i = \sum_{j \in \mathcal{I}(G) \cap \bar{N}_G(i)} B_{ji} \mathbf{u}_j \quad \forall i \in [n], \tag{1}$$

where $N_G(i) = \{j \in [n] \mid (i, j) \in E\}$ denotes the set of neighboring nodes of $i$ in $G$, and $\bar{N}_G(i) = N_G(i) \cup \{i\}$. Here $\mathbf{B} \in \mathbb{R}^{n \times \alpha}$ is a *coefficient matrix* that expresses $\mathbf{U}$ in terms of the bases features $\{\mathbf{u}_i\}_{i \in \mathcal{I}(G)}$. Note, we also assume $\mathbf{B}$ is such that any $\alpha \times \alpha$ submatrix of $\mathbf{B}$ is of rank $\alpha$, which ensures none of the dependent features can be represented as a linear combination of the other dependent features, or precisely $\mathcal{I}(G)$ is a maximal independent set of the independent nodes and all the dependent items $[n] \setminus \mathcal{I}(G)$, can only be represented as a unique linear combination of the independent nodes $\mathcal{I}(G)$. Thus we assume $B_{ij} = 0$ whenever $(i, j) \notin E$, and the subset of vectors in $\mathbf{U}$ corresponding to the items in the independent set $\mathcal{I}(G)$ are *linearly independent*. Thus $d \geq \alpha$, and $\{\mathbf{u}_i\}_{i \in \mathcal{I}(G)}$ form a basis for $span(\mathbf{U})$. Hence $B_{ij} = 1$, if $j = i$, or else $B_{ij} = 0$, $\forall j \neq i$, $\forall i \in [\alpha]$. We denote $\alpha = |\mathcal{I}(G)|$; clearly, it becomes the independent number of $G$ if $\mathcal{I}(G)$ corresponds to a maximum independent set. We will henceforth assume $\mathcal{I}(G) = [\alpha]$, w.l.o.g, unless specified otherwise. (see Fig. 1a for further illustration).

**Preference Model.** We introduce the *feature Bradley–Terry–Luce model* (f-BTL) where the probability of preferring item $i$ over $j$ is given by: $P_{ij} = \frac{e^{(\mathbf{w}^T \mathbf{u}_i)}}{e^{(\mathbf{w}^T \mathbf{u}_i)} + e^{(\mathbf{w}^T \mathbf{u}_j)}}$, $\mathbf{w} \in \mathbb{R}^d$. Note that the f-BTL model reduces to the standard BTL model when $\alpha = n$ and $\mathbf{U}$ is the standard basis. Clearly the '*score vector*' $\boldsymbol{\theta} \in \mathbb{R}^n$ for f-BTL model turns out to be $\theta_i = \mathbf{w}^T \mathbf{u}_i$.

**Sampling Model.** We assume that a set $M$ of $m \in [\binom{n}{2}]$ pairs is generated where each pair is chosen with some probability $p \in [0, 1]$. Each pair in $M$ is compared $K$ times independently according to f-BTL model.

**Remark.** (1) *shows that two items with similar set of neighbours in $\mathcal{I}(G)$ are similar in terms their features. This along with f-BTL model ensures two similar items are also similar in their scores $\theta_i$, and hence rankings.*

**Problem.** Under f-BTL model and given $\mathbf{U}$, for what values of $m$ and $K$ can one find an estimated score vector $\hat{\boldsymbol{\theta}}$ such that $P\left(\frac{\|\boldsymbol{\theta}-\hat{\boldsymbol{\theta}}\|_2}{\|\boldsymbol{\theta}\|_2} \leq \epsilon\right) > 1 - \delta$ ? Here $\epsilon > 0$ and $\delta \in [0, 1]$ are two given problem parameters, which respectively denote the allowable error limit and performance confidence.

**Performance Error.** The error above is measured in terms of *normalized $l_2$ error* $\frac{\|\boldsymbol{\theta}-\hat{\boldsymbol{\theta}}\|_2}{\|\boldsymbol{\theta}\|_2}$, which is a natural performance measure for score based probability models (e.g. BTL, Thurstone etc.) [16, 3]. Moreover, it is actually suitable for measuring ranking performances as it upper bounds the pairwise disagreement error – the weighted Kendall-Tau loss [16]:

$$pd(\boldsymbol{\theta}, \hat{\boldsymbol{\theta}}) = \left(\frac{1}{n\|\boldsymbol{\theta}\|_2^2} \sum_{i<j} (\theta_i - \theta_j)^2 \mathbf{1}\big((\theta_i - \theta_j)(\hat{\theta}_i - \hat{\theta}_j)<0\big)\right)^{\frac{1}{2}}$$

Thus giving a $(\epsilon, \delta)$ guarantee for the normalized $\ell_2$-error also ensures the same for $pd(\boldsymbol{\theta}, \hat{\boldsymbol{\theta}})$. We use both of the losses in our experiment evaluations (Section 6).

## 2.2 RELATED WORKS

Ranking from pairwise comparisons has been studied extensively in various disciplines owning to its huge practical importance, reviewing all lies beyond the scope of this work. We review only the works most relevant to our setting. The most related work is [17], however, they assume the features to lie in some low dimensional space and use a matrix completion-based approach to predict the ranking. Note that the low-rank assumption is a *global* assumption on the features that might miss out completely on the exact dependencies on the items. [8] also consider a feature preference information model, but do not analyze the graph theoretic aspects of feature dependencies.[9], [3] also use a least squares-based approach, but without any feature information. [16, 24, 6, 18, 22] [7], [19], [21] work in the pairwise ranking setting under different probabilistic models (including BTL model), but again none of them use features explicitly and hence are sub-optimal for our setting (as we will see in the experiments). [11] work in a setting where the probabilities come from some unknown low-dimensional feature embedding of the items. However, they require the pairs to be queried actively, whereas our work focuses on random (passive) selection of pairs. There is also a rich ranking literature on noisy sorting [5], approximation algorithms [2], dueling bandits [25] etc., which are fundamentally different from the passive setting under the BTL model considered here. Table 1 summarizes the sample complexities of a few related works.

Previous results show that under the standard BTL model, the *Rank Centrality* [16] [19], *MLE* under the BTL model [22] and the *Least Squares* [3] algorithms need $O(n \log n)$ comparisons to achieve a small error with probability at least $1 - O(poly(1/n))$. However, these algorithms do not con-

sider the features explicitly. The *Feature Low Rank* model of [17] uses features but requires $O(d^2 \log(n))$ pairs to be compared. Another related work is [12], which proposes an estimator for the parameters of a generalized linear parametric model, which includes classical preference models like Bradley-Terry and Thurstone. By addressing the violation of independence, they prove a sample complexity guarantee, showing that with Gaussian-distributed features, the estimator converges to a rescaled version of the model parameters based on the ambient dimension, number of samples, and comparisons. Their results indicate that achieving an accuracy $\epsilon > 0$ in model parameters requires $\Omega(dn \log^3 n/\epsilon^2)$ comparisons when the number of samples is $\Omega(d/\epsilon^2)$, which they validate through experiments on synthetic data. We show our proposed fBTL-LS algorithm requires only $O(\alpha \log \alpha)$ samples.

| Ranking Model | Sampling Technique | Sample Complexity |
|---|---|---|
| Noisy permutation [5] | Active | $O(n \log n)$ |
| Low $d$-dimensional embedding [11] | Active | $O(d \log^2 n)$ |
| Deterministic tournament [2] | Active | $O(n\text{poly}(\log n))$ |
| Rank-$r$ preference with $\nu$ incoherence [9] | Passive | $O(n\nu r(\log n)^2)$ |
| Bradley Terry Luce (BTL) [16] | Passive | $O(n \log n)$ |
| Noisy permutation [24] | Passive | $O(n \log n)$ |
| Low $r$-rank pairwise preference [19] | Passive | $O(nr \log n)$ |
| Low $d$-rank feature with BTL [17] | Passive | $O(d^2 \log n)$ |
| Rank aggregation balancing features [8] | Passive | $O(n)$ |
| **f-BTL** ($\alpha$ 'independent items') [This work] | Passive | $O(\alpha \log \alpha)$ |

Table 1: State-of-the-art vs Our work

## 3 ANALYSIS: KNOWN PREFERENCES

We begin by analyzing the problem for the noiseless case where for every pair $(i, j)$ that is compared, we have access to the exact value for $P_{ij}$. This analysis will shed light into the structure of the problem which will be useful later to analyse the case when $P_{ij}$s are unknown and need to be estimated from its noisy observations (Section 4). Under this setting, the goal is to bound the number of samples $m$ needed to *exactly* recover the score vector $\boldsymbol{\theta}$ where $\theta_i = \mathbf{w}^T \mathbf{u}_i \ \forall i \in [n]$. From Equation 1, we have that $\mathbf{w}^T \mathbf{u}_i = \sum_{j \in \mathcal{I}(G)} B_{ji} \mathbf{w}^T \mathbf{u}_j,$

or equivalently, $\quad \theta_i = \sum_{j \in \mathcal{I}(G)} B_{ji} \theta_j \ \forall i \in [n]. \quad (2)$

As we have access to $\mathbf{U}$ and $\mathbf{B}$, we only need to recover the scores of $\theta_j = \mathbf{w}^T \mathbf{u}_j \ \forall j \in [\alpha]$ so that the remaining scores can be computed using Equation 2. For a pair $(i, j)$, under the f-BTL model, the following holds:

$$\sum_{k=1}^{\alpha} \gamma_k^{ij} \theta_k = \sum_{k=1}^{\alpha} \gamma_k^{ij} (\mathbf{w}^T \mathbf{u}_k) = \log\left(\frac{P_{ij}}{P_{ji}}\right) \quad (3)$$

where $\gamma_k^{ij} = B_{ik} - B_{jk}$. Note that, from (1), this clearly implies $\gamma_k^{ij} = 0$ if $k \notin N(i) \cup N(j)$ as both $B(i, k) = B(j, k) = 0$ in that case. Eqn. (3) shows that knowing $P_{ij}$

for any pair $(i, j)$ gives rise to a linear equation involving the score vectors corresponding to the items only in $\mathcal{I}(G)$. Since the f-BTL model is invariant to constant shift of the score vector $\boldsymbol{\theta}$, we can w.l.o.g. assume that one of the item score to be 0 (with appropriate shift). Thus to recover the item scores, we only need $\mathcal{I}(G) - 1$ linearly independent equations of type Eqn. (3) that can be used to solve for the scores of the items in $\mathcal{I}(G)$, i.e $\{\theta_i\}_{i \in \mathcal{I}(G)}$. However, if the coefficient $\gamma_k^{ij}$ is 0 in (3) corresponding to the pair/edge $(i, j)$, then it does not involve $\theta_k$. *Thus, the equations of the selected pairs should be such that each item in $\mathcal{I}(G)$ appears in* at least *one of the equations so that it can be solved for.*

Thus our problem now is to compute the number of pairs needed to ensure that with high probability each item in $\mathcal{I}(G)$ appears in at least one equation of the form of Equation 3. To compute this number, we need to explicitly model the dependencies among features. We do this below and prove the necessary result using *the Hall's marriage theorem*, a classical result from graph matching theory. We state the theorem below for convenience.

**Hall's Marriage Theorem.** [10] Let $C = (A \cup A', E)$ be a finite bipartite graph and for any $S \subseteq A$, $N_C(S)$ denote the neighbours of $S$ in $A'$. Then $C$ admits a matching entirely covering $A$ if $|N_C(S)| \geq |S| \ \forall S \subseteq A$.

The bipartite graph $C = (A \cup A', \Delta)$ for our purpose is defined as follows: Set $A$ is just the set of items in the independent set i.e., $A = \mathcal{I}(G)$. (Recall $\mathcal{I}(G) = [\alpha]$). Set $A'$ consists of $\binom{n}{2}$ nodes, each corresponding to an edge $(i, j)$. For an edge $(i, j)$, define

$$F_{ij} = \{k \in \mathcal{I}(G) : \gamma_k^{ij} \neq 0\} \tag{4}$$

Thus $F_{ij}$ is a subset of independent nodes $\mathcal{I}(G)$ which are adjacent to at least either of item $i$ or $j$ (as otherwise $\gamma_k^{ij} = 0$, as argued above). Hence by observing the preference $P_{ij}$ of the pair $(i, j)$, we have an equation involving the items in $F_{ij}$. We define the edge set $\Delta$ such that an edge from node $k \in \mathcal{I}(G)$ to an edge $(i, j)$ is present in the bipartite graph $C$ iff $k \in F_{ij}$. For any set of edges $M \subseteq \binom{n}{2}$, define the reduced bipartite graph $C_M = (\mathcal{I}(G) \cup M, \Delta_M)$ by restricting the $A'$ to $M$ and defining $\Delta_M$ correspondingly. (see Fig. 1b).

**Theorem 3.1.** *Given a set of edges $M \subseteq \binom{n}{2}$, the bipartite graph $C_M = (\mathcal{I}(G) \cup M, \Delta_M)$ admits a matching that covers $A$ iff the system of linear equations induced by edges admits a unique solutions.*

Theorem 3.1 gives us a novel way to analyse the number of pairs needed to obtain enough (linearly independent) equations to uniquely solve for the score vector $\boldsymbol{\theta}$. In particular, we only need to bound the probability that the Hall's marriage condition is not met to get a bound on the number of pairs needed. (This is since when the

condition is met, a matching cover would give $\mathcal{I}(G)$ linearly independent equations to solve for the base scores of items in $\mathcal{I}(G)$, i.e $\{\theta_i\}_{i \in \mathcal{I}(G)}$). Before we prove the result, we need the following definitions for a given set $M$. Let $M_k$ denote the neighbours of node $k$ in $C_M$. Let $c_I = |\cup_{k \in I} M_k|, d_I = |\cap_{k \in I} M_k|, I \subseteq \mathcal{I}(G)$. We now prove the main result of this section:

**Theorem 3.2 (Bound On Error Probability).** *Given a relation graph $G$, feature matrix $\mathbf{U}$, a set of pairs $M$ where $|M| = m$ generated according to the sampling model above (where each pair is chosen with probability $p$), and the exact preference probabilities $P_{ij} \ \forall (i, j) \in M$, the probability that the score vector $\boldsymbol{\theta}$ is same as that estimated score vector $\hat{\boldsymbol{\theta}}$ that is got by solving the equations obtained is bounded by*

$$\mathbf{P}(\hat{\boldsymbol{\theta}} \neq \boldsymbol{\theta}) \leq \sum_{q=1}^{\min\{\alpha(G), d_{\max}(G)+1\}} \sum_{I \subseteq \mathcal{I}(G) \| |I|=q} \binom{d_I}{q-1} p^{q-1} (1-p)^{(c_I - (q-1))},$$

$d_{\max}(G)$ being the maximum degree of $G$.

*Proof.* **(sketch)** From Theorem 3.1 we have that one only fails to recover the true $\boldsymbol{\theta}$ if and only if the edge set $\Delta_M$ of the bipartite graph $C_M$ fails to cover $A$. Thus:

$$\mathbf{P}(\boldsymbol{\theta} \neq \hat{\boldsymbol{\theta}}) = \mathbf{P}(\{A \text{ is not covered by } C_M\})$$
$$= \mathbf{P}(\{\exists S' \subseteq A \text{ s.t. } |N_{C_M}(S')| < |S'|\}) \text{ (Hall's Marriage)}$$

Now if we denote the event $F_i := \{\exists S' \subseteq A \text{ s.t. } |S'| = i \text{ and } S' \text{ is not covered by } C_M\}, \forall i \in [\alpha(G)]$, and recalling $A = [\alpha(G)]$, one can further show

$$\mathbf{P}(\boldsymbol{\theta} \neq \hat{\boldsymbol{\theta}}) = \mathbf{P}(\{\exists S' \subseteq A \text{ s.t. } |N_{C_M}(S')| < |S'|\})$$
$$= \mathbf{P}(F_1 \cup F_2 \cup F_3 \ldots F_{\alpha(G)}) = \mathbf{P}(F_1)$$
$$+ \mathbf{P}(F_2 \cap F_1^c) + \ldots + \mathbf{P}(F_{\alpha(G)} \cap F_{\alpha(G)-1}^c) \tag{5}$$

Assuming the pairwise node preferences are drawn according to the edges sampled from an Erdős-Rényi random graph $\mathcal{G}(n, p)$ and applying Thm. 3.1 on the event $F_q \cap F_{q-1}^c$ for any $1 \leq q \leq \alpha(G)$, we get:

$$\mathbf{P}(F_q \cap F_{q-1}^c) = \mathbf{P}\big(\{\exists S' \subseteq A, |S'| = q, \ S' \text{ is not covered by } C_M \text{ and } \forall S_1' \subset A, |S_1'| < q, \ S_1' \text{ is covered by } C_M\})$$
$$\leq \sum_{I \subseteq \mathcal{I}(G) \| |I|=q} \binom{d_I}{q-1} p^{q-1} (1-p)^{c_I - q}, \tag{6}$$

where the last inequality follows from the observation that for any $S' \subseteq A, |S'| = q$ if $S'$ is not covered by $C_M$ but all it subsets $S_1' \subset S'$ are, then $\mathcal{G}(n, p)$ must have sampled exactly $q - 1$ edges from $\cap_{i \in S'} M_i$ and none from $\big( \cup_{i \in I} M_i \setminus \cap_{i \in I} M_i \big)$. Combining (5) in (6):

$$\mathbf{P}(\boldsymbol{\theta} \neq \hat{\boldsymbol{\theta}}) \leq P(F_1) + \ldots + P(F_{\alpha(G)} \cap F_{\alpha(G)-1}^c)$$
$$= \sum_{q=1}^{\alpha(G)} \sum_{I \subseteq \mathcal{I}(G) \| |I|=q} \binom{d_I}{q-1} p^{q-1} (1-p)^{c_I - (q-1)},$$

where we assume $\binom{x}{y} = 0$, if $x < y$. The result follows further noting that if $d_{\max}(G) < \alpha(G)$, then for any $I \subseteq [\alpha(G)]$ such that $|I| > (d_{\max} + 1)$, then $d_I = 0$. The complete proof is given in Appendix A.2. $\qquad\square$

**Remark.** The above theorem gives us a way of choosing $p$ such that the probability of not satisfying the Hall's condition (and hence not having enough equations to solve) can be bounded by a suitable value. As can be seen in the Theorem, the quantities of interest are $c_I$ and $d_I$ which capture the dependencies among the feature vectors of the nodes in the graph.

For several graphs, these quantities are easily computable, yielding the sample complexity bounds:

**Theorem 3.3** (**Sample Complexity for Common Graphs**). *Under the settings of Theorem 3.2, the sample complexity bounds for the following graphs are: 1. $m = O(n \log(\frac{n}{\delta}))$ for a disconnected graph, star graph, or cycle, 2. $m = O(\log(\frac{1}{\delta}))$ for a clique, 3. $m = O(r \log(\frac{r}{\delta}))$ for union of $r$ disconnected cliques.*

*Proof.* **(sketch)** The results could be obtained by first deriving the exact expression of $\mathbf{P}(\boldsymbol{\theta} \neq \hat{\boldsymbol{\theta}})$ for the specific graphs and solving for $p$ equating it to $\delta$. The required sample follows subsequently from the expected number of sampled edges $p\binom{n}{2}$. Eg., for $r$-Disconnected Cliques: Say $G$ has $r \in [n]$ disconnected cliques, $G_1, G_2, \ldots G_r$, each with $d \in [n]$ edges (i.e. for each $k \in [r]$, $|E(G_k)| = d$), assuming $n = rd$. Thus in this case $\alpha(G) = r$. Without loss of generality let $\mathcal{I}(G) = \{1, 2, \ldots r\}$. Then $\forall k \in [r]$, we have $M_k = \{(i,j) \mid (i,j) \in E(G_k)\} \cup \{(k,j) \mid j \in [n] \setminus \{k\}\}$. Thus $n_k = \binom{d}{2} + (r-1)$. Moreover note that $\forall I \subseteq [n]$, $|I| = 2$, $c_I = 2(\binom{d}{2} + (r-1)) - 1 = d(d-1) + (r-2)$, $d_I = 1$ and $|I| \geq 3$, $d_I = 0$.

Then applying Theorem 3.2 and noting $d_{\max}(G) \leq \lceil \frac{n}{r} \rceil$ one can get: $\mathbf{P}(\boldsymbol{\theta} \neq \hat{\boldsymbol{\theta}}) \leq r^2(e^{-p(\binom{d}{2}+r-1)})$. Now solving $r^2(e^{-p(\binom{d}{2}+r-1)}) \leq \delta$ this implies $p \geq \frac{1}{\binom{d}{2}+(r-1)} \log\left(\frac{r^2}{\delta}\right)$. Thus the expected number of edges (pairwise preferences) in the random graph required is atleast $p\binom{n}{2} = \frac{n(n-1)/2}{d(d-1)/2+r-1} \log\left(\frac{r^2}{\delta}\right) \geq \frac{n(n-1)r^2}{n(n-r)+2r^2(r-1)} \log\left(\frac{r^2}{\delta}\right) \geq r \log\left(\frac{r^2}{\delta}\right)$, where the last inequality follows assuming $r < \frac{n}{\sqrt{2}}$. Moreover setting $d = 1$ and $d = n$, we can recover the for disconnected and complete graphs respectively etc. The derivation for all the cases are in Appendix A.3. $\qquad\square$

**Remark.** Theorem 3.3 captures the connection between the structure of the relation graph $G([n], E)$ (induced by the features) and the sample complexity for recovering the item scores $\boldsymbol{\theta}$, under f-BTL model. E.g., if the graph is a clique, then there is only one independent vector and we need only

$O(1)$ pairwise samples; but for a disconnected graph, star or cycle where $\alpha = O(n)$, we recover the $O(n \log n)$ result for BTL model [16]. Moreover, there are graphs (e.g. $r$-disconnected cliques where $\alpha = r$) where the sample complexity scale as $O(\alpha \log(\alpha))$ (independent to $n$). Thus we get significant improvement in the sample complexity by exploiting the structure of the features which [17] fails to achieve. Sample complexities of few other graphs, e.g. regular graphs and trees are discussed in Appendix A.3.

It is also worth noting that the main structural assumption we exploited in Theorem 3.3 towards achieving the $O(\alpha \log \alpha)$ sample complexity is the low $\alpha$-dimensional embedding. Indeed, for a more general overview of our graph theoretic problem framework in (1), could assume $\mathcal{I}(G)$ to be an index set of some basis items, where the set $\{u_i \in \mathbb{R}^\alpha \mid i \in \mathcal{I}(G)\}$ represents a basis of the set of item features $\mathbf{U}$. Further, to mimic (1), now we assume a corresponding coefficient matrix $\tilde{\mathbf{B}}$ s.t. $\mathbf{U} = \tilde{\mathbf{B}}\mathbf{U}_\alpha$, where $\mathbf{U}_\alpha$ represents the "basis matrix" with vectors in $\{\mathbf{u}_i \mid i \in \mathcal{I}(G)\}$ stacked in the columns of $\mathbf{U}_\alpha$. In fact, note we do not need the knowledge of $\mathbf{U}_\alpha$ apriori: As given the true feature matrix $U$, we can derive one basis (by Gauss elimination or even Gram-Schmidt) that spans the feature space set $\mathbf{U}$. This is precisely what we adapted for our real-data experiments in Section 6.2.

# 4 GENERAL CASE: UNKNOWN PREFERENCES

In this section, we consider the original problem where we don't have access to the exact $P_{ij}$ values but only estimates of it available from the $K$ independent comparisons made. In this setting, we cannot expect to solve the linear equations exactly. We propose f-BTL, a least squares based algorithm, shown in Algorithm 1 to solve for the score vector. Let the graph induced by the edge set $M$ on the $n$ nodes be called the *comparison graph*. The node-edge incidence matrix $\mathbf{Q} \in \mathbb{R}^{n \times m}$ used in the algorithm is $\mathbf{Q}\mathbf{Q}^T$ which is the standard unnormalized Laplacian of the comparison graph i.e., $\mathbf{L} = \mathbf{Q}\mathbf{Q}^T = \mathbf{D} - \mathbf{A}$ where $\mathbf{D}$ is the diagonal matrix of degrees and $\mathbf{A}$ is the adjacency matrix. Algorithm 1 is motivated using the fact that when the true probabilities are known exactly, following holds:

$$\mathbf{Q}^T \mathbf{B} \mathbf{v} = \mathbf{y} \qquad (7)$$

where $\forall (i,j) \in M, y_{ij} = \log\left(\frac{P_{ij}}{P_{ji}}\right)$ and where $\mathbf{v} \in \mathbb{R}^\alpha$ such that $v_i = \theta_i \; \forall i \in [\alpha]$, $\mathbf{y} = (y_{ij})_{(i,j) \in M} \in \mathbb{R}^m$. Above relation simply follows as: $y_{ij} = \log\left(\frac{P_{ij}}{P_{ji}}\right) = \log\left(\frac{e^{\theta_i}}{e^{\theta_j}}\right) = \theta_i - \theta_j, \; \forall i, j \in [n]$ by the property of f-BTL model (Section 2.1). But since only noisy estimates $\hat{\mathbf{y}}$ are available instead of true $\mathbf{y}$, we take a least squares approach. The details is described in Algorithm 1.

## 4.1 CONNECTIVITY

The results of [3] show the sample complexity for the least squares algorithm for standard BTL model depends on how well connected the comparison graph is. Precisely, this is measured w.r.t the second Eigenvalue of the Laplacian $\mathbf{L}$ which is $0$ if and only if the comparison graph is disconnected. Thus when the comparison graph is disconnected, there is no way to recover the score vector in the standard BTL case. However, as we will see below, our analysis will depend on the least eigenvalue of the matrix $\tilde{\mathbf{Q}}\tilde{\mathbf{Q}}^T$ and not the Laplacian matrix. The important point to note here is that *even if the comparison graph is disconnected, the fBTL-LS algorithm may still recover the score vector*. This is because of the fact that the algorithm makes use of the matrix $B$ of coefficients to relate scores across possibly disconnected components in the comparison graph.

---

**Algorithm 1** Algorithm: fBTL-LS
---
**Require:** $G$, $\mathbf{U}$, a set $M$ of $m$ pairs each compared $K$ times.

   Compute $\mathbf{B}$ from $\mathbf{U}$ such that Equation 1 is satisfied for all $\mathbf{u}_i, i \in [n]$.

   Compute the node-edge incidence matrix $\mathbf{Q} \in \mathbb{R}^{n \times m}$ from $M$. Let $\tilde{\mathbf{Q}} = \mathbf{B}^T\mathbf{Q}$

   Compute $\hat{P}_{ij} = \begin{cases} \text{fraction of times } i \text{ beats } j \ \forall (i,j) \in M \\ 0 \quad \forall (i,j) \notin M \end{cases}$

   Compute $\hat{\mathbf{y}} \in \mathbb{R}^m$ where $\forall (i,j) \in M, \hat{y}_{ij} = \log\left(\frac{\hat{P}_{ij}}{\hat{P}_{ji}}\right)$

   Solve  $\hat{\mathbf{v}} = \arg\min_{\mathbf{x} \in \mathbb{R}^\alpha} \|\tilde{\mathbf{Q}}^T\mathbf{x} - \hat{\mathbf{y}}\|$

   Set $\hat{\theta}_i = \begin{cases} \hat{v}_i \ \forall i \in [\alpha] \\ \text{compute using Equation (2)} \ \forall i \notin [\alpha] \end{cases}$

   **return** score vector $\hat{\boldsymbol{\theta}}$

---

An example of this is shown in Figure 2. Here $n = 3$ and $M = \{(1,2),(1,3),(4,5)\}$ and $m = |M| = 3$. The comparison graph as can be seen in the figure is disconnected. The nodes circled in red are assumed to be the independent set nodes. The exact relation between the feature vectors of the independent set i.e., $\{\mathbf{u}_1, \mathbf{u}_2\}$ and those not in the independent set i.e., $\{\mathbf{u}_3, \mathbf{u}_4, \mathbf{u}_5\}$ are given by the matrix $\mathbf{B}$ shown in the figure. It can be verified for this example that the matrix $\mathbf{B}^T\mathbf{L}\mathbf{B}$ (also shown in the figure) has non zero eigenvalues though the Laplacian is block diagonal (which happens iff the comparison graph is disconnected).

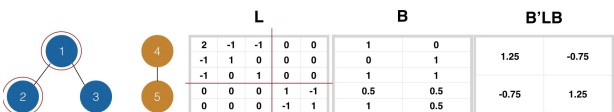

Figure 2: A disconnected comparison graph for which the $\mathbf{B}^T\mathbf{L}\mathbf{B}$ has non-zero minimum eigenvalue

**Theorem 4.1 (Recovery Guarantee for fBTL-LS Algo-**

**rithm).** *Let $M$ be a set of $m$ edges generated as per the sampling model and let each pair in $M$ be compared $K$ times independently according to the f-BTL model. Then for any positive scalar $K \geq 6(1 + e^{2b})^2 \log n$, with probability at least $1 - \frac{2m}{n^3}$, the normalized $\ell_2$-error of Algorithm 1 satisfies*

$$\frac{\|\hat{\boldsymbol{\theta}} - \boldsymbol{\theta}\|}{\|\boldsymbol{\theta}\|} \leq \frac{2}{a} \cdot \sqrt{\frac{\lambda_{\max}(\mathbf{B}^T\mathbf{B})}{\lambda_{\min}(\mathbf{B}^T\mathbf{B})}} \cdot \sqrt{\frac{m}{\alpha}} \cdot \frac{\sqrt{\lambda_n}}{\lambda_1},$$

$\lambda_1 = \min\{\lambda > 0 \mid \lambda \text{ is an eigen value of } \mathbf{B}^T\mathbf{L}\mathbf{B}\}$, $\lambda_n = \lambda_{\max}(\mathbf{B}^T\mathbf{L}\mathbf{B})$. $\lambda_{\min}(\mathbf{B}^T\mathbf{B})$ and $\lambda_{\max}(\mathbf{B}^T\mathbf{B})$ *respectively denotes the minimum and maximum non-zero eigenvalues of the positive semi-definite matrix $\mathbf{B}^T\mathbf{B}$. $a, b > 0$ denote the range of the f-BTL parameter such that $|\theta_i| \geq a, \forall i \in [\alpha]$ and $|\theta_i| \leq b, \forall i \in [n]$.*

*Proof.* **(sketch)** Let us denote the *reduced Laplacian* matrix by $\tilde{\mathbf{L}} = \tilde{\mathbf{Q}}\tilde{\mathbf{Q}}^T = \mathbf{B}^T\mathbf{Q}\mathbf{Q}^T\mathbf{B} = \mathbf{B}^T\mathbf{L}\mathbf{B}$ which is clearly positive semi-definite and has all non-negative eigenvalues. Let $f(\mathbf{x}) = \|\tilde{\mathbf{Q}}^T\mathbf{x} - \hat{\mathbf{y}}\|^2$, then note that $\hat{\mathbf{v}} = \arg\min_{\mathbf{x} \in \mathbb{R}^\alpha} f(\mathbf{x})$ in Algorithm 1 would satisfy the optimality condition $\nabla f(\hat{\mathbf{v}}) = 0$ when

$$\tilde{\mathbf{Q}}\hat{\mathbf{y}} = \tilde{\mathbf{Q}}\tilde{\mathbf{Q}}^T\hat{\mathbf{v}} = \tilde{\mathbf{L}}\hat{\mathbf{v}}, \tag{8}$$

On the other hand, assuming $\mathbf{v} \in \mathbb{R}^\alpha$ s.t. $v_i = \theta_i, \forall i \in [\alpha]$ and $\mathbf{y} \in \mathbb{R}^m$ be such that $y_{ij} = \log\left(\frac{P_{ij}}{P_{ji}}\right)$, we have $\mathbf{v} = \arg\min_{\mathbf{x} \in \mathbb{R}^\alpha} \|\tilde{\mathbf{Q}}^T\mathbf{x} - \mathbf{y}\|^2$ which gives

$$\tilde{\mathbf{Q}}\mathbf{y} = \tilde{\mathbf{L}}\mathbf{v}. \tag{9}$$

Above condition holds for any $i, j \in [n]$, $y_{ij} = \theta_i - \theta_j$, and so $\mathbf{y} = \mathbf{L}^T\boldsymbol{\theta} = \mathbf{L}^T\mathbf{B}\mathbf{v} = \tilde{\mathbf{Q}}^T\mathbf{v}$, where the second equality holds due to (2). Combining (8) and (9) we get $\tilde{\mathbf{Q}}(\mathbf{y} - \hat{\mathbf{y}}) = \tilde{\mathbf{L}}(\mathbf{v} - \hat{\mathbf{v}})$ from which it can be shown that, $\lambda_{\min}(\tilde{\mathbf{L}}\tilde{\mathbf{L}}^T)\|\mathbf{v} - \hat{\mathbf{v}}\|^2 \leq \lambda_{\max}(\tilde{\mathbf{Q}}^T\tilde{\mathbf{Q}})\|\mathbf{y} - \hat{\mathbf{y}}\|^2$. Noting $\lambda_{\max}(\tilde{\mathbf{Q}}^T\tilde{\mathbf{Q}}) = \lambda_{\max}(\tilde{\mathbf{Q}}\tilde{\mathbf{Q}}^T) = \lambda_n$ and $\lambda_{\min}(\tilde{\mathbf{L}}\tilde{\mathbf{L}}^T) = (\lambda_{\min}(\tilde{\mathbf{L}}))^2 = (\lambda_{\min}\tilde{\mathbf{Q}}\tilde{\mathbf{Q}}^T)^2 = \lambda_1^2$ above further implies:

$$\|\mathbf{v} - \hat{\mathbf{v}}\| \leq \frac{\|\mathbf{y} - \hat{\mathbf{y}}\|\sqrt{\lambda_n}}{\lambda_1}. \tag{10}$$

Now in order to bound $\|\mathbf{y} - \hat{\mathbf{y}}\| = \sqrt{\sum_{(i,j) \in E}(y_{ij} - \hat{y}_{ij})^2}$, we first note: $|y_{ij} - \hat{y}_{ij}| \leq |(\log P_{ij} - \log \hat{P}_{ij})| + |(\log P_{ji} - \log \hat{P}_{ji})|$. Denoting $\nu_{ij} = |P_{ij} - \hat{P}_{ij}|$ and applying *Hoeffding's Inequality*:

$$\mathbf{P}\left(\nu_{ij} \geq \eta\right) = \mathbf{P}\left(|P_{ij} - \hat{P}_{ij}| \geq \eta\right) \leq 2e^{-2\eta^2 K} \tag{11}$$

As $|\theta_i| \leq b, \forall i \in [n]$, we have $\frac{1}{1+e^{2b}} \leq P_{ij} \leq \frac{e^{2b}}{1+e^{2b}}, \forall i, j \in [n]$. Also as $K \geq 6(1 + e^{2b})^2 \log n$, using

(11), and further taking union bound over all pairs in $M$, we get with probability atleast $\left(1 - \frac{2m}{n^3}\right)$:

$$\mathbf{P}\left(\forall i, j \in [n], \nu_{ij} < \frac{P_{ij}}{2}\right) > \left(1 - \frac{2m}{n^3}\right). \qquad (12)$$

Define $g : [0, 1] \mapsto \mathbb{R}$, such that $g(p) = \log(p), \ \forall p \in [0, 1]$. Using Taylor's theorem, one can obtain a $p^* \in [P_{ij} - \nu_{ij}, P_{ij} + \nu_{ij}]$ such that

$$\log \hat{P}_{ij} = \log P_{ij} + \frac{1}{p^*}(\hat{P}_{ij} - P_{ij}), \text{ or equivalently,}$$

$$\frac{\log(\hat{P}_{ij}) - \log P_{ij}}{(\hat{P}_{ij} - P_{ij})} = \frac{1}{p^*} \leq \frac{2}{P_{ij}},$$

where the last inequality follows from (12) with probability at least $(1 - \frac{2m}{n^3})$. Furthermore, in the high probability event, as $|\hat{P}_{ij} - P_{ij}| < \frac{P_{ij}}{2}$, one can show $\|\mathbf{y} - \hat{\mathbf{y}}\| \leq 2\sqrt{m}$. Using this to (10) we get

$$\|\mathbf{v} - \hat{\mathbf{v}}\| \leq \frac{\|\mathbf{y} - \hat{\mathbf{y}}\|\sqrt{\lambda_n}}{\lambda_1} \leq \frac{2\sqrt{m\lambda_n}}{\lambda_1} \qquad (13)$$

with probability at least $\left(1 - \frac{1}{n}\right)$. The proof finally follows noting since $|\theta_i| \geq a, \ \forall i \in [\alpha]$, we have $\|\mathbf{v}\| \geq a\sqrt{\alpha}$. Moreover, as $\boldsymbol{\theta} = \mathbf{B}\mathbf{v}$, $\|\boldsymbol{\theta}\| \geq \sqrt{\lambda_{\min}(\mathbf{B}^T\mathbf{B})}\|\mathbf{v}\| \geq a\sqrt{\alpha\lambda_{\min}(\mathbf{B}^T\mathbf{B})}$. On the other hand, $\hat{\boldsymbol{\theta}} = \mathbf{B}\hat{\mathbf{v}}$ thus,

$$\|\boldsymbol{\theta} - \hat{\boldsymbol{\theta}}\| = \|\mathbf{B}(\mathbf{v} - \hat{\mathbf{v}})\| \leq \sqrt{\lambda_{\max}(\mathbf{B}^T\mathbf{B})}\|\mathbf{v} - \hat{\mathbf{v}}\|.$$

Combining above observations with (13) yields the desired bound. The proof is given in Appendix B.1. $\qquad \square$

**Remark.** Thm. 4.1 shows that the normalized error is bounded by a product of 4 terms. The first term $\frac{2}{a}$ can be treated as a constant that depends on the minimum score of the f-BTL model – a sensitivity component of the error bound. The second term is the condition number of the feature coefficient matrix $\mathbf{B}$ and captures how the features interact with each other. The third term depends on the number of pairs seen in $M$. When $|M| = m = \alpha \log \alpha$, this term becomes $\sqrt{\log \alpha}$. The fourth term grows depending on how many samples one sees as it depends on $L$ which is the Laplacian of the comparison graph. If both $\lambda_n$ and $\lambda_1$ are $O(\log \alpha)$, then the normalized error is a constant with probability at least $1 - poly(\frac{1}{n})$. Thus, the result essentially says that if one sees $O(\alpha \log \alpha)$ samples and $\mathbf{B}$ is such that both $\lambda_1$ and $\lambda_n$ are $O(\log \alpha)$, then the normalized error is bounded by a small constant. Thus the $m$ in the numerator could be misleading, as one expects decreasing performance error with increasing $m$. However as explained above, combining the effect of all $m$-dependent factors including eigenvalues of $B'LB$, the error bound on the right hand side decreases as $m$ scales as $O(\alpha \log \alpha)$.

# 5  LOWER BOUND

In this section, we show how the achievable $\ell_2$-error rate of the fBTL-LS algorithm (Theorem 4.1), compares to the minimax $\ell_2$-error rate possible, over the class of feature Bradley-Terry-Luce (f-BTL) model. Theorem 5.1 proves an information-theoretic lower bound for the $\ell_2$-error rate achievable by any learning algorithm for estimating the score parameters of the f-BTL model.

**Theorem 5.1 (Lower Bound for estimating the parameters of f-BTL model).** *Let us consider the following set of score vectors $\Theta_{\mathbf{B}}(a, b)$ of a f-BTL model defined with respect to the coefficient matrix $\mathbf{B}$ and range parameters $a, b > 0$ such that: $\mathbf{B}(a, b) = \{\theta \in \mathbb{R}^n \mid \theta \text{ satifies } (2), |\theta_i| \leq a \ \forall i \in [\alpha], |\theta_i| \geq b \ \forall i \in [n]\}$.*

*Now suppose the learner (an algorithm to estimate scores of a f-BTL model) is given access to noisy pairwise preferences sampled according to a $\mathcal{G}(n, p)$ Erdős-Rényi random graph with $p = \frac{\varsigma}{n}$ for some $\zeta > 0$, such that $K$ independent noisy pairwise preferences are available for each sampled pair, generated according to some unknown f-BTL model in $\Theta_{\mathbf{B}}(a, b)$. Then if $\hat{\boldsymbol{\theta}} \in \mathbb{R}^n$ be the learner's estimated f-BTL score vector based on the sampled pairwise preferences, upon which environment chooses a worst case true score vector $\boldsymbol{\theta} \in \Theta_{\mathbf{B}}(a, b)$, then for any such learning algorithm one can show that*

$$\sup_{\boldsymbol{\theta} \in \Theta_{\mathbf{B}}(a, b)} \frac{\mathbf{E}[\|\hat{\boldsymbol{\theta}} - \boldsymbol{\theta}\|]}{\|\boldsymbol{\theta}\|} \geq \frac{\sqrt{\lambda_{\min}(\mathbf{B}^T\mathbf{B})}}{16b\lambda_{\max}(\mathbf{B}^T\mathbf{B})\sqrt{448\zeta K e^{2(b+1)}}},$$

*the expectation is over the randomness of the algorithm.*

Our proof technique uses a constructive argument to generate the score vectors $\boldsymbol{\theta}$ from a uniform distribution that respects the f-BTL model in the dynamic range $|\theta_i| \in [a, b], \ \forall i \in [n]$, and solves the stochastic inference problem into a multi-way hypothesis testing problem. The full proof is given in Appendix C.1.

*Proof.* **(sketch)** We solve the above problem reducing it to a multi-class hypothesis testing problem as follows: Consider we are given a set of $N$ score vectors $\{\boldsymbol{\theta}^1, \boldsymbol{\theta}^2, \ldots \boldsymbol{\theta}^N\} \subset \Theta_B(a, b)$ s.t. $\|\boldsymbol{\theta}^{k_1} - \boldsymbol{\theta}^{k_2}\| \geq \delta$, for any two score vectors $\boldsymbol{\theta}^{k_1}, \boldsymbol{\theta}^{k_2}$ such that $k_1, k_2 \in [N]$. Then given the set of pairwise preferences generated by an unknown sore vector $\boldsymbol{\theta} = \boldsymbol{\theta}^L$ ($L$ being a random index selected uniformly $[N]$), the hypothesis testing task is to identify the index of the score vector $L$.

Now given any algorithm that predicts a score vector $\hat{\boldsymbol{\theta}}$ based on the given set of pairwise preferences from the f-BTL model $\boldsymbol{\theta}^L$, sampled according to a $\mathcal{G}(n, p)$ Erdős-Rényi random graph with $p = \frac{\varsigma}{n}$ for some $\zeta > 0$, such that $K$ independent noisy pairwise preferences are available for each sampled pair, one natural way to estimate $L$ is by $\hat{L} = \arg\min_{k \in [N]} \|\hat{\boldsymbol{\theta}} - \boldsymbol{\theta}^k\|$. Note that for $\hat{L}$ to be different

that $L$, it has to be the case that $\|\hat{\boldsymbol{\theta}} - \boldsymbol{\theta}\| \geq \frac{\delta}{2}$. Thus one can write $\mathbf{E}[\|\hat{\boldsymbol{\theta}} - \boldsymbol{\theta}\|] \geq \frac{\delta}{2}\mathbf{P}(\hat{L} \neq L)$. Further applying a similar information theoretic analysis as [16], one gets $\mathbf{E}[\|\hat{\boldsymbol{\theta}} - \boldsymbol{\theta}\|] \geq \frac{\delta}{2}\left[1 - \frac{\frac{K\zeta}{2N^2}\sum_{k_1 \in [N]}\sum_{k_2 \in [N]}\|e^{\boldsymbol{\theta}^{k_1}} - e^{\boldsymbol{\theta}^{k_2}}\|^2 + \log 2}{\log N}\right]$

Thus the remaining task is to construct a set of $N$ score vectors $\{\boldsymbol{\theta}^1, \boldsymbol{\theta}^2, \ldots \boldsymbol{\theta}^N\} \subset \Theta_B(a, b)$ which are well separated, so to get suitable bounds on the terms $\|e^{\boldsymbol{\theta}^{k_1}} - e^{\boldsymbol{\theta}^{k_2}}\|^2$, $\forall k_1, k_2 \in [N]$ to obtain the desired lower bound for which we carefully constructed the score vectors as follows: For any $k \in [N]$, we construct the $k^{th}$ score vector $\theta^k$ set of the set of $N$ random score vectors as follows: **1.** Draw $\alpha$ many random variables $X_1^k, X_2^k, \ldots X_\alpha^k \sim$ Unif$\left[\left(\frac{1}{2} - \beta\delta\right), \left(\frac{1}{2} + \beta\delta\right)\right]$, where $\beta$ is a constant to be adjusted later. **2.** Set $\theta_i^k = a + (b - a)X_i^k$, $\forall i \in [\alpha]$, $0 < a < b < 1$. **3.** Consider the coefficient matrix $\mathbf{B} \in \mathbb{R}_+^{n \times \alpha}$ such that $\sum_{j=1}^\alpha B_{ij} = 1$, $\forall i \in [n]$. **4.** Set the remaining score vectors $\theta_i^k$ according to (2) for all $i \in [n] \setminus [\alpha]$. The claim now follows proving the following two lemmas

**Lemma 5.2.** $\frac{1}{6}(b-a)^2\alpha\beta^2\delta^2 \leq \|\boldsymbol{\theta}_{[\alpha]}^{k_1} - \boldsymbol{\theta}_{[\alpha]}^{k_2}\|^2 \leq \frac{7}{6}(b-a)^2\alpha\beta^2\delta^2$, *for all* $k_1, k_2 \in [N] \times [N]$, *with probability at least* $\left(1 - N^2 e^{-\frac{\alpha}{32}}\right)$,

**Lemma 5.3.** *Given any two* $\boldsymbol{\theta}, \boldsymbol{\theta}' \in [a, b]^n$, *such that* $0 < a < b < 1$ $\|e^{\boldsymbol{\theta}} - e^{\boldsymbol{\theta}'}\|^2 \leq e^{2(b+1)}\|\boldsymbol{\theta} - \boldsymbol{\theta}'\|^2$

which combined with the above derived lower bound on $\mathbf{E}[\|\hat{\boldsymbol{\theta}} - \boldsymbol{\theta}\|]$ yields the result. The complete proof can be found in Appendix C.1. $\square$

**Remark.** *Since* $m = \binom{n}{2}p$, *or equivalently* $\zeta = pn = O\left(\frac{m}{n}\right)$, *the above bound suggests that the left hand side is bounded by a small constant upon observing* $m = \alpha \log \alpha$ *pairs for* $K \geq \frac{n}{\alpha \log \alpha}$ *– which exactly matches our derived upper bound of Thm. 4.1 for any* $n \geq \alpha \log \alpha \log n$, *establishing tightness of the results.*

# 6 EXPERIMENTS

We compared our algorithm *fBTL-LS* with three state-of-the-art methods: (i) *Ordinary Least Squares (OLS )* [3], (ii) *Rank Centrality (RC)* [16] and (iii) *Inductive Pairwise Ranking* based on inductive matrix completion (*IMC*) [17]. The first two algorithms do not use any feature information while the third algorithm does.

**Performance Measures: 1. Normalized $\ell_2$-error:** For experiments where there is a true score vector, we use the normalized $\ell_2$ error between the estimated score vector and the true score vector $\left(\frac{\|\hat{\boldsymbol{\theta}} - \boldsymbol{\theta}\|}{\|\boldsymbol{\theta}\|}\right)$.

**2. Pairwise disagreement (pd) error:** Suppose $\mathbf{P}^* \in [0, 1]^{n \times n}$ denotes the pairwise preference matrix corresponding to the true (and unknown) score $\theta$, given by $P_{ij}^* = \frac{e^{\theta_i}}{e^{\theta_i} + e^{\theta_j}}$ and $\hat{\mathbf{P}} \in [0, 1]^{n \times n}$ be the estimated

preference matrix return by the algorithm (note if the algorithm returns a score vector estimate $\hat{\boldsymbol{\theta}}$, we compute $\hat{P}_{ij} = \frac{e^{\hat{\theta}_i}}{e^{\hat{\theta}_i} + e^{\hat{\theta}_j}}$ $\forall i, j \in [n]$), then pd-error is defined as: $\text{pd}(\hat{\mathbf{P}}, \mathbf{P}^*) = \frac{2}{n(n-1)}\sum_{i<j}\left(\mathbb{I}(\hat{P}_{ij} \geq 0.5 \wedge P_{ij}^* < 0.5) + \mathbb{I}(\hat{P}_{ij} < 0.5 \wedge P_{ij}^* > 0.5)\right)$

**Remark.** *This counts the fraction of pairs where* $\mathbf{P}^*$ *and* $\hat{\mathbf{P}}$ *disagree – the* Kendall's Tau ranking loss *[14, 16] between true and estimated ranking (* $\boldsymbol{\theta}$ *and* $\hat{\boldsymbol{\theta}}$ *).*

**3. Sample complexity($\text{sc}(\epsilon)$):** Minimum number of pairwise comparisons required to be observed to obtain normalized $\ell_2$-error $\left(\frac{\|\hat{\boldsymbol{\theta}} - \boldsymbol{\theta}\|}{\|\boldsymbol{\theta}\|}\right) < \epsilon$.

## 6.1 EXPERIMENTS ON SYNTHETIC DATASETS

We consider three different settings— *Type-I plots*: with increasing node size ($n$), *Type-II plots*: with increasing sampling rate ($p$) but fixed node size ($n$) and independence number ($\alpha$) *Type-III plots*: with increasing independence number ($\alpha$) and fixed node size ($n$).

**Graphs used.** We use 3 different graphs for synthetic experiments: $(1)$ $r$-disconnected cliques: Union of $r$-cliques $(2)$ $d$-regular graphs: Graphs with each node having degree $d$ and $(3)$ $k$-ary trees: Trees with every node having $k$-children (except the leaf nodes).

**Data generation** For each of the above type of graphs $G$, we first fix a maximum independent set $\mathcal{I}(G)$ of $G$, and embed the $i^{th}$ node of $\mathcal{I}(G)$ with the $i^{th}$ canonical basis vector of $\mathbb{R}^\alpha$, i.e. $\mathbf{u}_i = \mathbf{e}_i$, $\forall i \in [\alpha]$. Thus our feature dimension is $d = \alpha$. We next generate a random coefficient matrix $\mathbf{B} \in \mathbb{R}^{n \times \alpha}$ and obtain the feature embedding $\{\mathbf{u}_i\}_{i=1}^n \subset \mathbb{R}^\alpha$ of rest of the items using (1). Choose a random vector $\mathbf{w} \in \mathbb{R}^\alpha$ and assign a BTL score $\theta_i = \mathbf{w}^T\mathbf{u}_i$ to every node $i \in [n]$ as defined in (2). Finally $\boldsymbol{\theta} = (\theta_1, \theta_2, \ldots, \theta_n)$ is normalized to $\ell_2$-norm 1, i.e. $\|\boldsymbol{\theta}\|_2 = 1$, setting $\theta_i = \frac{\theta_i}{\|\boldsymbol{\theta}\|_2}$, $\forall i \in [n]$.

**Parameter setting.** As follows from the data generation, the feature dimension $d$ is equal to the independence number $\alpha = |\mathcal{I}(G)|$ of $G$ in each case. We also fix $K = 1000$ (unless performance is reported against $K$), and report the average performances over 50 runs.

**TYPE-I PLOTS: INCREASING NODE SIZE ($n$)**

We compare the algorithms, with varying node size ($n$), on three different graphs: (1) Union of 10 disconnected cliques on $n$ nodes, (2) $d$-Regular graph of $n$ nodes with fixed degree $d = 10$ and (3) Full binary tree of $n$ nodes. The results are reported in Figure 3. They clearly reflect the superior performance of *fBTL-LS* for each of the three performance measures.

**Results.** For 10-disconnected cliques, $\alpha = 10$ is fixed for all $n$, unlike graph (2) and (3) where $\alpha$ scales with $n$. The sam-

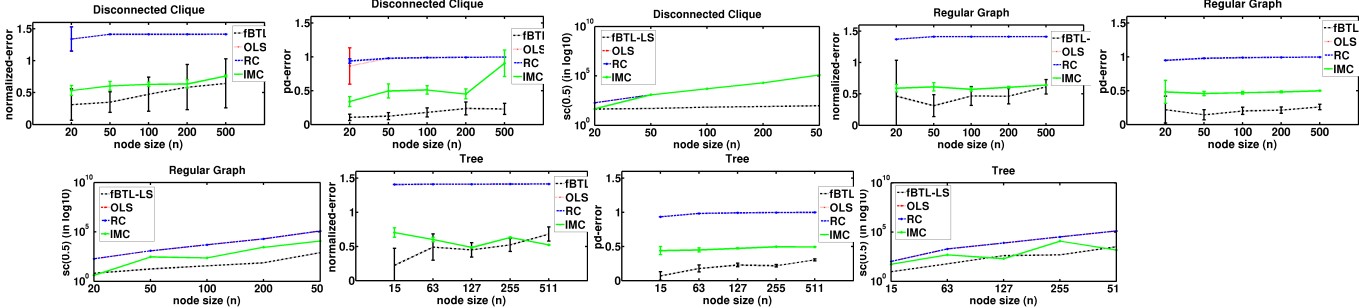

Figure 3: Performance vs $n$ on (1) 10-disconnected clique, (2) 10-regular graph (3) full binary tree

ple complexity sc(0.5) of *fBTL-LS* for achieving a target error $\epsilon = 0.5$ for 10-disconnected clique is almost constant for $n$ from 20 to 500, unlike the rest of the algorithms where it scales with $n$–which justifies our claim of the required sample complexity to be $O(\alpha \log \alpha)$, as also remarked in Thm. 4.1. This also justifies why for 10-regular graph and full binary tree, the sample complexity of *fBTL-LS* monotonically increases with $n$, as $\alpha$ itself scales with $n$ for them.

**Remark.** *The above reflects how our algorithm finds the position of a newly added item in an already estimated ranking without collecting extra pairwise preferences, as long as it lies in the span of $\mathcal{I}(G)$ (i.e. $\alpha$ remains fixed), the sample complexity remains unaffected too (e.g. 10-disconnected cliques) – This is a significant advantage of our method over the rest which cannot exploit the underlying item dependencies and thus needs to observe preference information of the newly added nodes leading to increased sample complexity with increasing $n$.*

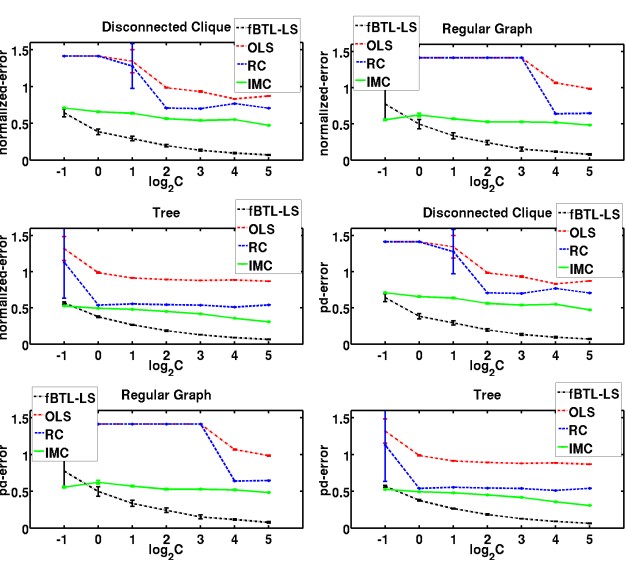

Figure 4: Performance vs $p$, where $p = \frac{C\alpha \log \alpha}{\binom{n}{2}}$ on (1) 100-disconnected clique, (2) 50-regular graph and (3) full binary tree, with $\sim 500$ nodes in each

**TYPE-II PLOTS: INCREASING $p$, FIXED $n, \alpha$**

Here we compare the algorithms with varying sampling rate $p$ for two estimation error metrics, normalized $\ell_2$-error and pairwise disagreement pd($\hat{\mathbf{P}}, \mathbf{P}^*$), on the following three different graphs: (1) Union of 100 disconnected cliques on 500 nodes, i.e. each clique of 5 nodes, (2) 50-Regular graphs on 500 nodes, each of degree $d = 50$ and (3) Full binary tree of height 8 (511 nodes). Thus in each case, $n$ and $\alpha$ are kept fixed, with $p$ to be set as $p = \frac{C\alpha \log \alpha}{\binom{n}{2}}$, $C$ varying from 0.5 to 32.

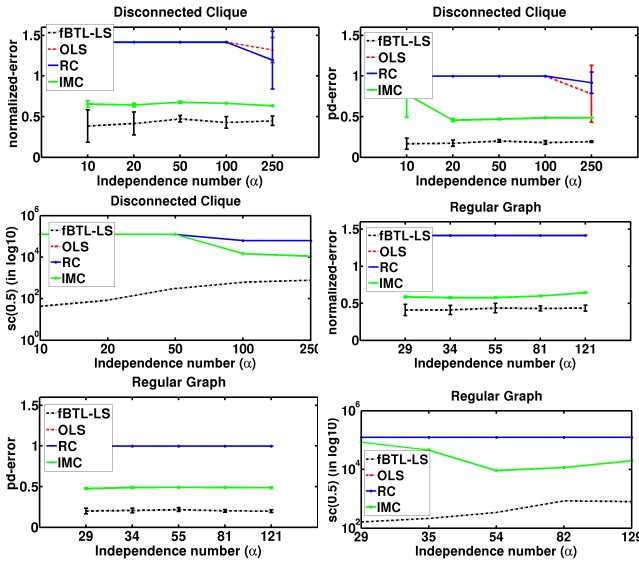

Figure 5: Performance vs $\alpha$ on disconnected cliques and $d$-regular graph ($n = 500$ nodes in each)

**Results.** Fig. 4 shows, as expected, the performance of all the algorithms gets improved with a higher sampling rate $p$. However, the performance improvement rate is far more drastic for *fBTL-LS* compared to the rest due to its inherent ability to exploit the feature correlation, and thus attains accurate score estimates faster.

**TYPE-III PLOTS: INCREASING $\alpha$ AND FIXED $n$**

In the third setup, we compare the four algorithms with varying independence set size (or independence number)

$\alpha$ for a fixed set of $n = 500$ nodes on the following two graphs: (1) Union of $r$-disconnected cliques over 500 nodes with varying $r$ and (2) $d$-Regular graph of 500 nodes with varying degree $d$ (Figure 5).

**Results.** The results show varying $p$ as $p = \frac{10(\alpha \log \alpha)}{\binom{n}{2}}$ normalized $\ell_2$-error and pd$(\hat{\mathbf{P}}, \mathbf{P}^*)$, remains almost constant validating the claim of the required sample complexity of *fBTL-LS* to be $O(\alpha \log \alpha)$, as follows from Theorem 4.1. The sample complexity curves on the other hand validate the dependency of sc$(0.5)$ on $\alpha$, which increases with higher values of $\alpha$, as expected.

## 6.2 REAL DATA EXPERIMENTS

We finally evaluate the algorithms on two benchmark real-world preference learning datasets: *car* and *sushi*.

**1. Car Dataset.** ([1]) It contains pairwise preferences of 20 cars given by 60 users, where each car is represented by a 6-dimensional feature vector. **2. Sushi Dataset.** ([13]) This dataset contains over 100 sushis rated according to their preferences, each sushi is represented by a 7-dimensional feature vector.

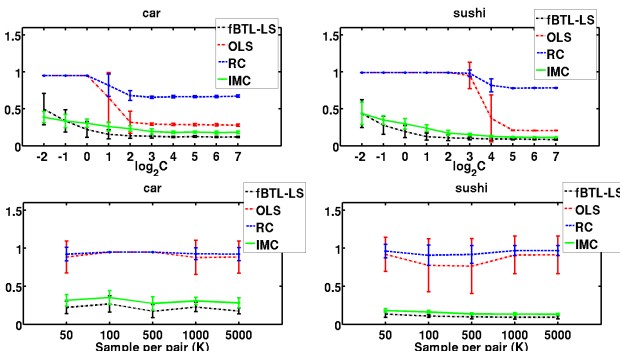

Figure 6: Pairwise disagreement error pd$(\hat{\mathbf{P}}, \mathbf{P}^*)$ vs sampling rate $\left( p = \frac{C\alpha \log \alpha}{\binom{n}{2}} \right)$, and number of repeated samples $(K)$ on *Car* and *Sushi*

**Setup.** Note that the real-world datasets do not satisfy any preference modeling assumption, e.g. BTL assumption, and hence there is no true score vector $\boldsymbol{\theta}$ associated to the item preferences. From the user preferences, we first compute the underlying pairwise preference matrix $\mathbf{P}^*$, where $P_{ij}^*$ is computed by taking the empirical average of number of times an item $i$ is preferred over item $j$. Further to construct the feature matrix $\mathbf{U}$, we use the provided feature information of the item set, that is provided in each dataset. Specifically, if each item is represented by $d$-dimensional feature vector (as described before, $d = 6$ for *Car* and $d = 7$ for *Sushi*), we find a set of $d$ items whose corresponding features are linearly independent that forms a basis of $\mathbb{R}^d$ and

use these $d$ items as the independent set $\mathcal{I}$. The coefficient matrix $\mathbf{B}$ is then constructed by representing the rest of the items as a linear combination of $\mathcal{I}$, such that it satisfies (1) (see Sec. 2.1).

**Performance Measure.** As noted above, the real-world datasets do not satisfy the BTL assumption, so there is no true score vector $\boldsymbol{\theta}$ associated with the item preferences. We however measure the performances of the algorithms with respect to the true preference matrix $\mathbf{P}^*$, using pairwise disagreement error pd$(\hat{\mathbf{P}}, \mathbf{P}^*)$.

In both cases, our algorithm outperforms the rest. We also evaluate the algorithms with an increasing number of repeated samples per pair $(K)$. As expected, it shows higher $K$ leads to improved performance (Fig. 6).

## 7 CONCLUSION AND FUTURE WORKS

Many of the state-of-the-art existing ranking algorithms either fail to utilize this feature information, or make broad low-rank assumptions that cannot capture the item dependencies through their corresponding feature representations. We introduce a feature-based probabilistic preference model, f-BTL, and propose a least squares-based algorithm, *fBTL-LS*, which is shown to achieve much tighter sample complexity for the problem of ranking from pairwise comparisons with feature information.

We have proposed a least squares-based algorithm and have shown theoretical recovery guarantees for the same. While least square-based algorithms are a natural choice, it would be interesting to see how Markov chain-based approaches, e.g. *Rank Centrality* [16] can be extended to accommodate feature information. One can also potentially consider the contextual setting introducing user features in addition. Analyzing the sample complexity for recovering partial ordering (e.g. top-K items) would be useful too.

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

# SUPPLEMENTARY: A GRAPH THEORETIC APPROACH FOR PREFERENCE LEARNING WITH FEATURE INFORMATION

## A  SUPPLEMENTARY FOR SECTION 3

### A.1  PROOF OF THEOREM 3.1

**Theorem 3.1.** *Given a set of edges $M \subseteq \binom{n}{2}$, the bipartite graph $C_M = (\mathcal{I}(G) \cup M, \Delta_M)$ admits a matching that covers $A$ iff the system of linear equations induced by edges admits a unique solutions.*

*Proof.* If there is a matching that covers $A$, then each node $i$ in $\mathcal{I}(G)$ has a distinct representative edge in $M$ which induces an equation containing $i$. Thus there are at least $\mathcal{I}(G)$ equations with each node appearing in at least one of them and hence the system can be solved for. More over the solution would be unique since these $\mathcal{I}(G)$ many induced equations would be linearly independent. It is important note in this regard that than all the equations (of form Eqn. (3)) emerges from any pair $(i, j)$ is would lead to a *linearly independent* equation–this is since we also assume $\mathbf{B}$ is such that any $\alpha \times \alpha$ submatrix of $\mathbf{B}$ is of rank $\alpha$ (see Sec. 2.1), which ensures none of the dependent features can be represented as a linear combination of the other dependent features. Above is crucial for the correctness of proof as it ensures all the linear equations induced through these cover-matching edges are linearly independent.

On the other hand, if there is no matching that covers $A$, then by Hall's marriage theorem [10], there must exist some subset $S \subseteq A$ such that it's neighbours $|N_{C_M}(S)| < |S|$. As the total number of equations that involve nodes in $S$ are less than the number of nodes, this set of equations cannot be solved for. $\qquad\square$

### A.2  PROOF OF THEOREM 3.2

**Theorem 3.2** (**Bound On Error Probability**)**.** *Given a relation graph $G$,feature matrix $\mathbf{U}$, a set of pairs $M$ where $|M| = m$ generated according to the sampling model above (where each pair is chosen with probability $p$), and the exact preference probabilities $P_{ij} \ \forall (i, j) \in M$, the probability that the score vector $\boldsymbol{\theta}$ is same as that estimated score vector $\hat{\boldsymbol{\theta}}$ that is got by solving the equations obtained is bounded by*

$$\mathbf{P}(\hat{\boldsymbol{\theta}} \neq \boldsymbol{\theta}) \leq \sum_{q=1}^{\min\{\alpha(G), d_{\max}(G)+1\}} \sum_{I \subseteq \mathcal{I}(G)||I|=q} \binom{d_I}{q-1} p^{q-1}(1-p)^{(c_I - (q-1))},$$

*Proof.* Note from Theorem 3.1 we have that one only fails to recover the true $\boldsymbol{\theta}$ if and only if the edge set $\Delta_M$ of the bipartite graph $C_M$ fails to cover $A$. Thus we have

$$\begin{aligned}
\mathbf{P}(\boldsymbol{\theta} \neq \hat{\boldsymbol{\theta}}) &= \mathbf{P}(\{A \text{ is not covered by } C_M\}) \\
&= \mathbf{P}(\{\exists S' \subseteq A \text{ s.t. } |N_{C_M}(S')| < |S'|\}) \quad \text{(by Hall's Marriage Theorem)}
\end{aligned}$$

We use $N_G(i)$ to denote the set of neighbours of node $i \in [n]$ in a graph $G$ and $\bar{N}_G(i)$ to denote the set of neighbours of node $i \in [n]$ in $G$ including $i$ itself, i.e. $\bar{N}_G(i) = N_G(i) \cup \{i\}$. Define $N_G(S) = \cup_{i \in S} N_G(i)$, $\forall S \subseteq V(G)$ and $\bar{N}_G(ij) = \left(\bar{N}_G(i) \cup \bar{N}_G(j)\right) \cap \mathcal{I}(G)$. Thus we can associate every node $k \in \mathcal{I}(G) = [\alpha(G)]$ in the independent set to a set of edges $M_k$ such that $(i, j) \in M_k \iff k \in \bar{N}_G(ij)$. Let us also denote $n_k = |M_k|$ and let $n_{\min} = \min_{\{k \in [\alpha(G)]\}} n_k$. More generally we denote $n_I = |\cap_{i \in I} M_i|$, $\forall I \subseteq [\alpha(G)]$.

We will also find it convenient to define $c_I = |\cup_{i \in I} M_i|$ and $d_I = |\cap_{i \in I} M_i|$, $\forall I \subseteq [\alpha(G)]$. Clearly when $|I| = 1$, say $I = \{i\}$, $i \in [n]$, $c_I = d_I = n_i$. In general, for $|I| = q$, $1 \leq q \leq \alpha(G)$ we have $c_I = \sum_{x=1}^{q} \sum_{J \subseteq I||J|=x} (-1)^{x-1} d_J$, where the size of the intersecting sets $d_I$s depends on specific the structure of the graph $G$ (see Theorem 3.3 for graph specific analysis).

Now if we denote the event $F_i := \{\exists S' \subseteq A \text{ s.t. } |S'| = i \text{ and } S' \text{ is not covered by } C_M\}$, $\forall i \in [\alpha(G)]$, and recalling $A = [\alpha(G)]$, we further get

$$\mathbf{P}(\boldsymbol{\theta} \neq \hat{\boldsymbol{\theta}}) = \mathbf{P}(\{\exists S' \subseteq A \text{ s.t. } |N_{C_M}(S')| < |S'|\})$$
$$= P(F_1 \cup F_2 \cup F_3 \ldots F_{\alpha(G)})$$
$$= P\big(F_1 \cup (F_2 \cap F_1^c) \cup (F_3 \cap F_2^c) \cup \ldots \cup (F_{\alpha(G)} \cap F_{\alpha(G)-1}^c)\big)$$
$$= P(F_1) + P(F_2 \cap F_1^c) + \ldots + P(F_{\alpha(G)} \cap F_{\alpha(G)-1}^c) \tag{14}$$

Assuming the pairwise node preferences are drawn according to the edges sampled from an Erdős-Rényi random graph $\mathcal{G}(n,p)$ and applying Theorem 3.1 on the event $F_1$, it is easy to see that

$$\mathbf{P}(F_1) = \mathbf{P}(\{\exists S' \subseteq A \text{ s.t. } |N_{C_M}(S')| < |S'| = 1\})$$

$$= \mathbf{P}\Big(\{\exists S' = \{k\}, \ k \in [\alpha(G)] \text{ s.t. no edge from } M_k \text{ is sampled in } \mathcal{G}(n,p)\}\Big) \leq \sum_{i=1}^{\alpha(G)} (1-p)^{n_i},$$

where the last inequality follows taking union bound over all singletons in $A = [\alpha(G)]$. Note that one can further bound above as $\mathbf{P}(F_1) \leq \alpha(G) \exp(-p n_{\min})$. In general, for any $1 \leq q \leq \alpha(G)$, one can similarly derive

$$\mathbf{P}(F_q \cap F_{q-1}^c)$$
$$= \mathbf{P}\big(\{\exists S' \subseteq A, |S'| = q, \ S' \text{ is not covered by } C_M \text{ and } \forall S_1' \subset A, |S_1'| < q, \ S_1' \text{ is covered by } C_M\}\big)$$
$$\leq \sum_{I \subseteq \mathcal{I}(G) || I| = q} \binom{d_I}{q-1} p^{q-1} (1-p)^{c_I - q}, \tag{15}$$

where the last inequality follows from the crucial observation that for any $S' \subseteq A$, $|S'| = q$ if $S'$ is not covered by $C_M$ but all it subsets $S_1' \subset S'$ are, then $\mathcal{G}(n,p)$ must have sampled exactly $q-1$ edges from $\cap_{i \in S'} M_i$ and none from $\big(\cup_{i \in I} M_i \setminus \cap_{i \in I} M_i\big)$. Using (15) in (14) we finally get,

$$\mathbf{P}(\boldsymbol{\theta} \neq \hat{\boldsymbol{\theta}}) \leq P(F_1) + P(F_2 \cap F_1^c) + \ldots + P(F_{\alpha(G)} \cap F_{\alpha(G)-1}^c)$$
$$= \sum_{q=1}^{\alpha(G)} \sum_{I \subseteq \mathcal{I}(G) || I| = q} \binom{d_I}{q-1} p^{q-1} (1-p)^{c_I - (q-1)},$$

where we assume $\binom{x}{y} = 0$, if $x < y$. Further note that if $d_{\max}(G) < \alpha(G)$, then for any $I \subseteq [\alpha(G)]$ such that $|I| > (d_{\max} + 1)$, we have $d_I = 0$, using which we further get

$$\mathbf{P}(\boldsymbol{\theta} \neq \hat{\boldsymbol{\theta}}) \leq \sum_{q=1}^{\min\{\alpha(G), \ (d_{\max}(G)+1)\}} \sum_{I \subseteq \mathcal{I}(G) || I| = q} \binom{d_I}{q-1} p^{q-1} (1-p)^{c_I - (q-1)}$$

Thus the claim follows. $\qquad\square$

## A.3 PROOF OF THEOREM 3.3

**Theorem 3.3** (**Sample Complexity for Common Graphs**). *Under the settings of Theorem 3.2, the sample complexity bounds for the following graphs are: 1. $m = O(n \log(\frac{n}{\delta}))$ for a disconnected graph, star graph, or cycle, 2. $m = O(\log(\frac{1}{\delta}))$ for a clique, 3. $m = O(r \log(\frac{r}{\delta}))$ for union of $r$ disconnected cliques.*

*Proof.* We will now analyse Theorem 3.2 for certain specific class of graphs. We will be using the same notations used in proof of Theorem 3.2 for the purpose.

1. **Fully Disconnected Graph:** Note that in this case $\alpha(G) = n$. Also note that $\forall k \in [n]$, $M_k = \{(k, i) \mid i \in [n] \setminus \{k\}\}$. Thus $n_k = n - 1$. Moreover $\forall I \subseteq [n], |I| = 2, c_I = 2n - 3, d_I = 1$, and if $|I| \geq 3, d_I = 0$.

   Now applying Theorem 3.2 and noting $d_{\max}(G) = 0$, we further get that,

$$\mathbf{P}(\boldsymbol{\theta} \neq \hat{\boldsymbol{\theta}}) \leq \sum_{q=1}^{\min\{\alpha(G),\,(d_{\max}(G)+1)\}} \sum_{I||I|=q} \binom{d_I}{q-1} p^{q-1}(1-p)^{c_I - q}$$

$$= \sum_{i=1}^{n}(1-p)^{n-1} + \sum_{i<j} p(1-p)^{2n-3-1}$$

$$= n(1-p)^{n-1} + \binom{n}{2} p(1-p)^{2n-4}$$

$$\leq n(e^{-p})^{n-1} + \frac{n(n-1)}{2} p(e^{-p})^{2n-4}$$

$$\leq n^2(e^{-p(n-1)})$$

$$\leq \delta,$$

   solving which we get $p \geq \frac{1}{(n-1)} \log\left(\frac{n^2}{\delta}\right)$. Thus the expected number of edges (pairwise preferences) in the random graph required is atleast $p\binom{n}{2} \geq \frac{n}{2} \log\left(\frac{n}{\delta}\right)$, which recovers the result for the usual BTL model.

2. **Complete Graph:** In this case $\alpha(G) = 1$. Without loss of generality assuming $\mathcal{I}(G) = \{1\}$, thus we have $M_1 = \{(i, j) \mid i, j \in [n]\}$. Thus $n_1 = \binom{n}{2}$. Moreover $\forall I \subseteq [n], |I| \geq 2, d_I = 0$.

   Applying Theorem 3.2 as before and noting $d_{\max}(G) = n$, we further get,

$$\mathbf{P}(\boldsymbol{\theta} \neq \hat{\boldsymbol{\theta}}) \leq \sum_{q=1}^{\min\{\alpha(G),\,(d_{\max}(G)+1)\}} \sum_{I||I|=q} \binom{d_I}{q-1} p^{q-1}(1-p)^{c_I - q}$$

$$= (1-p)^{\binom{n}{2}}$$

$$= (e^{-p})^{\binom{n}{2}}$$

$$\leq \delta,$$

   solving which one gets $p \geq \frac{1}{\binom{n}{2}} \log\left(\frac{1}{\delta}\right)$. Thus the expected number of edges (pairwise preferences) in the random graph required is atleast $p\binom{n}{2} \geq \log\left(\frac{1}{\delta}\right)$, which is intuitive as well since in a complete graph one needs the knowledge of only $\Omega(1)$ pairwise preferences to recover the exact ranking (i.e. $\boldsymbol{\theta}$) with high probability $(1 - \delta)$.

3. $r$-**Disconnected Cliques:** Say $G$ has exactly $r \in [n]$ disconnected cliques, $G_1, G_2, \ldots G_r$, each with $d \in [n]$ edges (i.e. for each $k \in [r]$, $|E(G_k)| = d$), assuming $n = rd$. Thus in this case $\alpha(G) = r$. Without loss of generality assume $\mathcal{I}(G) = \{1, 2, \ldots r\}$. Then $\forall k \in [r]$, we have $M_k = \{(i, j) \mid (i, j) \in E(G_k)\} \cup \{(k, j) \mid j \in [n] \setminus \{k\}\}$. Thus $n_k = \binom{d}{2} + (r - 1)$. Moreover $\forall I \subseteq [n], |I| = 2, c_I = 2(\binom{d}{2} + (r - 1)) - 1 = d(d - 1) + (r - 2), d_I = 1$ and $|I| \geq 3, d_I = 0$.

   Then applying Theorem 3.2 as above and noting $d_{\max}(G) \leq \lceil \frac{n}{r} \rceil$, we further get,

$$\mathbf{P}(\boldsymbol{\theta} \neq \hat{\boldsymbol{\theta}}) \leq \sum_{q=1}^{\min\{\alpha(G),\,(d_{\max}(G)+1)\}} \sum_{I\,||I|=q} \binom{d_I}{q-1} p^{q-1}(1-p)^{c_I-q}$$

$$= \sum_{i=1}^{r}(1-p)^{\binom{d}{2}+r-1} + \sum_{i<j,\,i,j\in[r]} p(1-p)^{d(d-1)+(r-2)-1}$$

$$= r(1-p)^{\binom{d}{2}+r-1} + \binom{r}{2}p(1-p)^{d(d-1)+(r-3)}$$

$$\leq r(e^{-p})^{\binom{d}{2}+r-1} + \frac{r(r-1)}{2}p(e^{-p})^{d(d-1)+(r-3)}$$

$$\leq r(r-1)(e^{-p(\binom{d}{2}+r-1)})$$

$$\leq r^2(e^{-p(\binom{d}{2}+r-1)}) \leq \delta,$$

solving which one can derive $p \geq \frac{1}{\binom{d}{2}+(r-1)}\log\left(\frac{r^2}{\delta}\right)$. Thus the expected number of edges (pairwise preferences) in the random graph required is atleast $p\binom{n}{2} = \frac{n(n-1)/2}{d(d-1)/2+r-1}\log\left(\frac{r^2}{\delta}\right) \geq \frac{n(n-1)r^2}{n(n-r)+2r^2(r-1)}\log\left(\frac{r^2}{\delta}\right) \geq r\log\left(\frac{r^2}{\delta}\right)$, where the last inequality follows assuming $r < \frac{n}{\sqrt{2}}$. Note that setting $d=1$ and $d=n$, one can recover the earlier bounds we derived for disconnected and complete graphs respectively.

4. **Star:** Note that in this case the size of the maximal independent set $\alpha(G) = (n-1)$. Without loss of generality assume $\mathcal{I}(G) = [n] \setminus \{1\}$. Thus we have that for any $k \in \mathcal{I}(G)$, $E_k = \{(k,j) \mid j \in [n] \setminus \{k\}\} \cup \{(1,j) \mid j \in [n] \setminus \{1\}\}$. Thus $n_k = (n-1)+(n-2) = 2n-3$. Moreover $\forall I \subseteq [n]$, $|I| = 2$, $d_I = (n-2)+1 = n-1$ and $c_I = 2(2n-3)-(n-1) = 3n-5$. For $|I| \geq 3$, $d_I = n-2$ and $c_I = (2n-3)|I| - (n-1)(\binom{|I|}{2}) + (n-2)(\binom{|I|}{3}) - \ldots + (-1)^{|I|-1}(n-2)$, e.g. when $|I| = 3$, $c_I = 4n-14$ etc.

Applying Theorem 3.2 as before and noting $d_{\max}(G) = (n-1)$, we further get,

$$\mathbf{P}(\boldsymbol{\theta} \neq \hat{\boldsymbol{\theta}}) \leq \sum_{q=1}^{\min\{\alpha(G),\,(d_{\max}(G)+1)\}} \sum_{I\,||I|=q} \binom{d_I}{q-1} p^{q-1}(1-p)^{c_I-q}$$

$$= (n-1)\left((1-p)^{2n-3} + \frac{n(n-2)}{2}p(1-p)^{2n-4}\right)$$

$$+ \binom{n-1}{3}\binom{n-2}{2}p^2(1-p)^{3n-12} + \ldots$$

$$\leq n^2(e^{-p(n-1)})$$

$$\leq \delta.$$

Similar to the case of *fully disconnected graph*, solving $p$ from above one can get that the expected number of edges (pairwise preferences) in the random graph required is atleast $p\binom{n}{2} = \left(\frac{n}{2}\log\left(\frac{n}{\delta}\right)\right)$.

5. **Cycle:** We will assume that $n = 2n' \geq 4$ is even, similar analysis can be done for the odd number of nodes as well. Thus in this case $\alpha(G) = n'$. Without loss of generality assume $\mathcal{I}(G) = \{2i \in [n] \mid i \in [n]\}$. Thus we have that for any $k \in \mathcal{I}(G)$, $E_k = \{(k,j) \mid j \in [n] \setminus \{k\}\} \cup \{(k-1,j) \mid j \in [n] \setminus \{k-1\}\} \cup \{((k+1) \mod k, j) \mid j \in [n] \setminus \{(k+1) \mod k\}\}$. Thus $n_k = (n-1)+(n-2)+(n-3) = 3(n-2)$. Moreover $\forall I \subseteq [n]$, $|I| = 2$, $d_I = (n-2)+1 = n-1$ and $c_I = 2(3n-6)-(n-1) = 2n-5$. For $|I| \geq 3$, $d_I = 0$.

Further applying Theorem (3.2) and noting $d_{\max}(G) = 2$, we further get,

$$\mathbf{P}(\boldsymbol{\theta} \neq \hat{\boldsymbol{\theta}}) \leq \sum_{q=1}^{\min\{\alpha(G),\,(d_{\max}(G)+1)\}} \sum_{I||I|=q} \binom{d_I}{q-1} p^{q-1}(1-p)^{c_I-q}$$

$$= \sum_{i=1}^{n'} (1-p)^{3n-6} + \sum_{I \subset \mathcal{I}(G), |I|=2} (n-1)p(1-p)^{2n-5-(n-1)}$$

$$= \frac{n}{2}(1-p)^{3n-6} + \frac{n(n-1)(n-2)}{8}p(1-p)^{n-4}$$

$$\leq \frac{n}{2}(1-p)^{3n-6} + \frac{n-2}{4}(1-p)^{n-4} \quad \left(\text{as } p\binom{n}{2} \geq 1\right)$$

$$\leq n(1-p)^{n-4}$$

$$\leq \delta,$$

solving which one can derive $p = f(\delta) \geq \frac{1}{n-4}\log\left(\frac{n}{\delta}\right)$. Thus the expected number of edges (pairwise preferences) in the random graph required is atleast $p\binom{n}{2} = \frac{n(n-1)/2}{n-4}\log\left(\frac{n}{\delta}\right) \geq \frac{n}{2}\log\left(\frac{n}{\delta}\right)$.

6. **$K$-ary Tree:** Let $h$ be the height of the tree and 1 denotes the root node. For any node $i \in [n]$, $par(i)$ and $ch(i)$ respectively denotes the parent and child nodes $i$. We will consider only trees of even height for the purpose, it is easy to derive a similar analysis for trees of odd height. Note that $n = (1 + K + K^2 + \ldots + K^h) = \frac{K^{h+1}-1}{K-1}$. Clearly the maximum independent set contains all the nodes which which are at a even length distance from the root, including the root itself. Thus $\alpha(G) = (1 + K^2 + K^4 + \ldots + K^h) = \frac{K^{h+2}-1}{K^2-1}$.

Note that for any $k \in \mathcal{I}(G)$, $N_G(k) \cap \mathcal{I}(G)^c = \{par(k) \cup ch(k)\}$. Also every node in $[n] \setminus \{\mathcal{I}(G)\}$ form $C = (K + K^3 + K^5 + \ldots + K^{h-1}) = \frac{K^{h+2}-1}{K^2-1}$ clusters, we denote them by $H_1, H_2, \ldots H_C$, such that for any $i \in [n] \setminus \{\mathcal{I}(G)\}$, $H_i = \{j \in \mathcal{I}(G) \mid j \in par(i) \cup ch(i)\}$. Thus $|H_i| = K + 1$. We will also abbreviate $N_G(\cdot)$ as $N(\cdot)$ for ease of notations.

Thus for any $k \in \mathcal{I}(G)$, $E_k = \{(k,j) \mid j \in [n] \setminus \{k\}\} \cup_{k' \in par(k) \cup ch(k)} \{(k',j) \mid j \in [n] \setminus \{k'\}\}$. This gives that $n_k = (n-1) + \sum_{i=2}^{K+2}(n-i) = (k+2)\frac{2n-k-3}{2}$. Moreover, for any $I \subseteq \mathcal{I}(G), |I| = 2$,

$$d_I = \begin{cases} 1 + (n-2), & \forall i,j \in I, |N(i) \cap N(j)| = 1, \\ 1, & otherwise, \end{cases}$$

$$c_I = \begin{cases} (K+1)(2n-K-2), & \forall i,j \in I, |N(i) \cap N(j)| = 1, \\ (K+2)(2n-k-3) - 1, & otherwise, \end{cases}$$

for any $I \subseteq \mathcal{I}(G), 3 \geq |I| \leq k+1$,

$$d_I = \begin{cases} (n-2), & \forall i,j \in I, |N(i) \cap N(j)| = 1, \\ 0, & otherwise, \end{cases}$$

$$c_I = \begin{cases} |I|(K+2)\frac{(2n-K-3)}{2} - \binom{|I|}{2}(n-1)\ldots(-1)^{|I|-1}(n-2), \\ \qquad \forall i,j \in I, |N(i) \cap N(j)| = 1, \\ |I|(K+2)\frac{(2n-K-3)}{2} - \binom{|I|}{2}1, & otherwise, \end{cases}$$

and for any $I \subseteq \mathcal{I}(G), |I| > K + 1, d_I = 0$. Now applying Theorem 3.2 as before and noting $d_{\max}(G) = K$ we further get,

$$\mathbf{P}(\boldsymbol{\theta} \neq \hat{\boldsymbol{\theta}}) \leq \sum_{q=1}^{\min\{\alpha(G),\,(d_{\max}(G)+1)\}} \sum_{I||I|=q} \binom{d_I}{q-1} p^{q-1}(1-p)^{c_I-q}$$

$$= \alpha(G)(1-p)^{\frac{(K+2)(2n-K-3)}{2}}+$$

$$C\binom{K+1}{2}(n-1)p(1-p)^{(K+1)(2n-K-2)-(n-1)}$$

$$\left(\binom{n}{2} - C\binom{K+1}{2}\right)p(1-p)^{(K+2)(2n-K-3)-1}$$

$$\sum_{K'=3}^{K+1} C\binom{K+1}{3}(n-2)p^2(1-p)^{"C_I"-2}$$

$$\leq \delta.$$

Unlike the previous cases this does not reduce to any non-trivial closed form upper bound of $p$ for deriving a generalized sample complexity bound for any $K$-ary tree, however one might use above to get sample complexities for some specific choices of $h$ and $K$.

$\square$

# B  SUPPLEMENTARY FOR SECTION 4

## B.1  PROOF OF THEOREM 4.1

**Theorem 4.1** (**Recovery Guarantee for fBTL-LS Algorithm**). *Let $M$ be a set of $m$ edges generated as per the sampling model and let each pair in $M$ be compared $K$ times independently according to the f-BTL model. Then for any positive scalar $K \geq 6(1+e^{2b})^2 \log n$, with probability at least $1 - \frac{2m}{n^3}$, the normalized $\ell_2$-error of Algorithm 1 satisfies*

$$\frac{\|\hat{\boldsymbol{\theta}} - \boldsymbol{\theta}\|}{\|\boldsymbol{\theta}\|} \leq \frac{2}{a} \cdot \sqrt{\frac{\lambda_{\max}(\mathbf{B}^T\mathbf{B})}{\lambda_{\min}(\mathbf{B}^T\mathbf{B})}} \cdot \sqrt{\frac{m}{\alpha}} \cdot \frac{\sqrt{\lambda_n}}{\lambda_1},$$

$\lambda_1 = \min\{\lambda > 0 \mid \lambda \text{ is an eigen value of } \mathbf{B}^T\mathbf{L}\mathbf{B}\}$, $\lambda_n = \lambda_{\max}(\mathbf{B}^T\mathbf{L}\mathbf{B})$. $\lambda_{\min}(\mathbf{B}^T\mathbf{B})$ *and* $\lambda_{\max}(\mathbf{B}^T\mathbf{B})$ *respectively denotes the minimum and maximum non-zero eigenvalues of the positive semi-definite matrix* $\mathbf{B}^T\mathbf{B}$. $a, b > 0$ *denote the range of the f-BTL parameter such that* $|\theta_i| \geq a$, $\forall i \in [\alpha]$ *and* $|\theta_i| \leq b$, $\forall i \in [n]$.

*Proof.* Let us denote the *reduced Laplacian* matrix by $\tilde{\mathbf{L}} = \tilde{\mathbf{Q}}\tilde{\mathbf{Q}}^T$. $\mathbf{L} = \mathbf{Q}\mathbf{Q}^T$ being the original graph Laplacian, the *reduced Laplacian* is given by $\tilde{\mathbf{L}} = \mathbf{B}^T\mathbf{Q}\mathbf{Q}^T\mathbf{B} = \mathbf{B}^T\mathbf{L}\mathbf{B}$ which is clearly positive semi-definite and has all non-negative eigenvalues. Define $f(\mathbf{x}) = \|\tilde{\mathbf{Q}}^T\mathbf{x} - \hat{\mathbf{y}}\|^2$. Note that $\hat{\mathbf{v}} = \arg\min_{\mathbf{x} \in \mathbb{R}^\alpha} f(\mathbf{x})$ in Algorithm 1 would satisfy the optimality condition $\nabla f(\hat{\mathbf{v}}) = 0$ when

$$\tilde{\mathbf{Q}}\hat{\mathbf{y}} = \tilde{\mathbf{Q}}\tilde{\mathbf{Q}}^T\hat{\mathbf{v}} = \tilde{\mathbf{L}}\hat{\mathbf{v}}, \tag{16}$$

On the other hand, assuming $\mathbf{v} \in \mathbb{R}^\alpha$ to be such that $v_i = \theta_i$, $\forall i \in [\alpha]$ and $\mathbf{y} \in \mathbb{R}^m$ be such that $y_{ij} = \log\left(\frac{P_{ij}}{P_{ji}}\right)$, we have $\mathbf{v} = \arg\min_{\mathbf{x} \in \mathbb{R}^\alpha} \|\tilde{\mathbf{Q}}^T\mathbf{x} - \mathbf{y}\|^2$ which gives

$$\tilde{\mathbf{Q}}\mathbf{y} = \tilde{\mathbf{L}}\mathbf{v}. \tag{17}$$

The above optimality condition holds as for any $i, j \in [n]$, $y_{ij} = \theta_i - \theta_j$, and so $\mathbf{y} = \mathbf{L}^T\boldsymbol{\theta} = \mathbf{L}^T\mathbf{B}\mathbf{v} = \tilde{\mathbf{Q}}^T\mathbf{v}$, where the second equality holds due to (2). Thus combining (16) and (17), we get

$$\tilde{\mathbf{Q}}(\mathbf{y} - \hat{\mathbf{y}}) = \tilde{\mathbf{L}}(\mathbf{v} - \hat{\mathbf{v}})$$

which further gives,

$$(\mathbf{y} - \hat{\mathbf{y}})^T \tilde{\mathbf{Q}}^T \tilde{\mathbf{Q}}(\mathbf{y} - \hat{\mathbf{y}}) = \|\tilde{\mathbf{Q}}(\mathbf{y} - \hat{\mathbf{y}})\|^2 = \|\tilde{\mathbf{L}}(\mathbf{v} - \hat{\mathbf{v}})\|^2 = (\mathbf{v} - \hat{\mathbf{v}})^T \tilde{\mathbf{L}}\tilde{\mathbf{L}}^T (\mathbf{v} - \hat{\mathbf{v}}),$$

from which we get

$$\lambda_{\min}(\tilde{\mathbf{L}}\tilde{\mathbf{L}}^T)\|\mathbf{v} - \hat{\mathbf{v}}\|^2 \leq \|\tilde{\mathbf{L}}(\mathbf{v} - \hat{\mathbf{v}})\|^2 = \|\tilde{\mathbf{Q}}(\mathbf{y} - \hat{\mathbf{y}})\|^2 \leq \lambda_{\max}(\tilde{\mathbf{Q}}^T \tilde{\mathbf{Q}})\|\mathbf{y} - \hat{\mathbf{y}}\|^2 \tag{18}$$

where $\lambda_{\min}(\tilde{\mathbf{L}}\tilde{\mathbf{L}}^T)$ is the smallest non-zero eigenvalue of the positive semi-definite matrix $(\tilde{\mathbf{L}}\tilde{\mathbf{L}}^T)$ and $\lambda_{\max}(\tilde{\mathbf{Q}}^T \tilde{\mathbf{Q}})$ being the largest eigenvalue of $(\tilde{\mathbf{Q}}^T \tilde{\mathbf{Q}})$. Now from standard results on matrix eigenvalues, we know that the set of non-zero eigenvalues of $\tilde{\mathbf{Q}}^T \tilde{\mathbf{Q}}$ and $\tilde{\mathbf{Q}}\tilde{\mathbf{Q}}^T$ are exactly same, which implies $\lambda_{\max}(\tilde{\mathbf{Q}}^T \tilde{\mathbf{Q}}) = \lambda_{\max}(\tilde{\mathbf{Q}}\tilde{\mathbf{Q}}^T) = \lambda_n$. Moreover, $\lambda_{\min}(\tilde{\mathbf{L}}\tilde{\mathbf{L}}^T) = (\lambda_{\min}(\tilde{\mathbf{L}}))^2 = (\lambda_{\min}\tilde{\mathbf{Q}}\tilde{\mathbf{Q}}^T)^2 = \lambda_1^2$. Thus from Equation 18, we get

$$\|\mathbf{v} - \hat{\mathbf{v}}\| \leq \frac{\|\mathbf{y} - \hat{\mathbf{y}}\|\sqrt{\lambda_n}}{\lambda_1}. \tag{19}$$

Now in order to bound $\|\mathbf{y} - \hat{\mathbf{y}}\| = \sqrt{\sum_{(i,j)\in E}(y_{ij} - \hat{y}_{ij})^2}$, first recall from the definition of $y_{ij}$ that $y_{ij} = \log\left(\frac{P_{ij}}{P_{ji}}\right) = \log P_{ij} - \log P_{ij}$, for any edge $(i,j) \in M$. Similarly we have $\hat{y}_{ij} = \log \hat{P}_{ij} - \log \hat{P}_{ij}$. Thus we have,

$$|y_{ij} - \hat{y}_{ij}| = |(\log P_{ij} - \log \hat{P}_{ij}) - (\log P_{ji} - \log \hat{P}_{ji})|$$
$$\leq |(\log P_{ij} - \log \hat{P}_{ij})| + |(\log P_{ji} - \log \hat{P}_{ji})| \tag{20}$$

Let us denote $\nu_{ij} = |P_{ij} - \hat{P}_{ij}|$. Clearly $|P_{ji} - \hat{P}_{ji}| = \nu_{ij}$ since $P_{ij} + P_{ji} = \hat{P}_{ij} + \hat{P}_{ji} = 1$. Note that the random variable $\hat{P}_{ij}$ is the average of $K$ samples from Bernoulli$(P_{ij})$, applying *Hoeffding's Inequality* we get

$$\mathbf{P}\left(\nu_{ij} \geq \eta\right) = \mathbf{P}\left(|P_{ij} - \hat{P}_{ij}| \geq \eta\right) \leq 2e^{-2\eta^2 K} \tag{21}$$

Now since $|\theta_i| \leq b$, $\forall i \in [n]$, we have $\frac{1}{1+e^{2b}} \leq P_{ij} \leq \frac{e^{2b}}{1+e^{2b}}$, $\forall i,j \in [n]$. Also as $K \geq 6(1+e^{2b})^2 \log n$, using (21) we further have

$$\mathbf{P}\left(\nu_{ij} \geq \frac{P_{ij}}{2}\right) \leq \mathbf{P}\left(\nu_{ij} \geq \frac{1}{2(1+e^{2b})}\right) \leq \frac{2}{n^3}, \quad \forall i,j \in [n] \tag{22}$$

Above thus implies that $\nu_{ij} = |P_{ij} - \hat{P}_{ij}| < \frac{P_{ij}}{2}$ with high probability of at least $(1 - \frac{2}{n^3})$, for $K = 6\log n(1+e^{2b})^2$. Further since $\nu_{ij} = \nu_{ji}$, using union bound over all pairs in $M$, we get that (22) holds true for all pairs $(i,j) \in [n]$ with probability atleast $\left(1 - \frac{2m}{n^3}\right)$, i.e.

$$\mathbf{P}\left(\forall i,j \in [n], \nu_{ij} < \frac{P_{ij}}{2}\right) > \left(1 - \frac{2m}{n^3}\right).$$

Define $g : [0,1] \mapsto \mathbb{R}$, such that $g(p) = \log(p)$, $\forall p \in [0,1]$. Using Taylor's theorem, one can obtain a $p^* \in [P_{ij} - \nu_{ij}, P_{ij} + \nu_{ij}]$ such that

$$\log \hat{P}_{ij} = \log P_{ij} + \frac{1}{p^*}(\hat{P}_{ij} - P_{ij}), \text{ or equivalently,}$$

$$\frac{\log(\hat{P}_{ij}) - \log P_{ij}}{(\hat{P}_{ij} - P_{ij})} = \frac{1}{p^*} \leq \frac{2}{P_{ij}},$$

where the last inequality follows from (22) with probability at least $(1 - \frac{2m}{n^3})$.

Furthermore, in the high probability event, as $|\hat{P}_{ij} - P_{ij}| < \frac{P_{ij}}{2}$ Thus we have

$$|\log(\hat{P}_{ij}) - \log P_{ij}| \leq 1, \quad \forall i,j \in [n].$$

combining above with (20) we get

$$|y_{ij} - \hat{y}_{ij}| \leq 2,$$

which implies $\|\mathbf{y} - \hat{\mathbf{y}}\| \le 2\sqrt{m}$. Applying above to (19) we thus get

$$\|\mathbf{v} - \hat{\mathbf{v}}\| \le \frac{\|\mathbf{y} - \hat{\mathbf{y}}\|\sqrt{\lambda_n}}{\lambda_1} \le \frac{2\sqrt{m\lambda_n}}{\lambda_1} \tag{23}$$

with probability at least $\left(1 - \frac{1}{n}\right)$. Finally note that since $|\theta_i| \ge a$, $\forall i \in [\alpha]$, we have $\|\mathbf{v}\| \ge a\sqrt{\alpha}$. Moreover, as $\boldsymbol{\theta} = \mathbf{B}\mathbf{v}$, $\|\boldsymbol{\theta}\| = \|\mathbf{B}\mathbf{v}\| \ge \sqrt{\lambda_{\min}(\mathbf{B}^T\mathbf{B})}\|\mathbf{v}\| \ge a\sqrt{\alpha\lambda_{\min}(\mathbf{B}^T\mathbf{B})}$. On the other hand, we have set $\hat{\boldsymbol{\theta}} = \mathbf{B}\hat{\mathbf{v}}$ thus,

$$\|\boldsymbol{\theta} - \hat{\boldsymbol{\theta}}\| = \|\mathbf{B}(\mathbf{v} - \hat{\mathbf{v}})\| \le \sqrt{\lambda_{\max}(\mathbf{B}^T\mathbf{B})}\|\mathbf{v} - \hat{\mathbf{v}}\|.$$

Combining above with (23), we finally have

$$\frac{\|\boldsymbol{\theta} - \hat{\boldsymbol{\theta}}\|}{\|\boldsymbol{\theta}\|} \le \frac{2\sqrt{m\lambda_n\lambda_{\max}(\mathbf{B}^T\mathbf{B})}}{a\lambda_1\sqrt{\alpha\lambda_{\min}(\mathbf{B}^T\mathbf{B})}},$$

with probability at least $\left(1 - \frac{2m}{n^3}\right)$ and the claim follows. $\qquad\square$

# C  SUPPLEMENTARY FOR SECTION 5

## C.1  PROOF OF THEOREM 5.1

**Theorem 5.1** (**Lower Bound for estimating the parameters of f-BTL model**). *Let us consider the following set of score vectors* $\Theta_{\mathbf{B}}(a, b)$ *of a f-BTL model defined with respect to the coefficient matrix* $\mathbf{B}$ *and range parameters* $a, b > 0$ *such that:* $\mathbf{B}(a, b) = \{\theta \in \mathbb{R}^n \mid \theta \text{ satifies (2)}, |\theta_i| \le a\ \forall i \in [\alpha], |\theta_i| \ge b\ \forall i \in [n]\}.$

*Now suppose the learner (an algorithm to estimate scores of a f-BTL model) is given access to noisy pairwise preferences sampled according to a* $\mathcal{G}(n, p)$ *Erdős-Rényi random graph with* $p = \frac{\zeta}{n}$ *for some* $\zeta > 0$*, such that* $K$ *independent noisy pairwise preferences are available for each sampled pair, generated according to some unknown f-BTL model in* $\Theta_{\mathbf{B}}(a, b)$*. Then if* $\hat{\boldsymbol{\theta}} \in \mathbb{R}^n$ *be the learner's estimated f-BTL score vector based on the sampled pairwise preferences, upon which environment chooses a worst case true score vector* $\boldsymbol{\theta} \in \Theta_{\mathbf{B}}(a, b)$*, then for any such learning algorithm one can show that*

$$\sup_{\boldsymbol{\theta} \in \Theta_{\mathbf{B}}(a,b)} \frac{\mathbf{E}[\|\hat{\boldsymbol{\theta}} - \boldsymbol{\theta}\|]}{\|\boldsymbol{\theta}\|} \ge \frac{\sqrt{\lambda_{\min}(\mathbf{B}^T\mathbf{B})}}{16b\lambda_{\max}(\mathbf{B}^T\mathbf{B})\sqrt{448\zeta Ke^{2(b+1)}}},$$

*the expectation is over the randomness of the algorithm.*

*Proof.* We solve the above problem reducing it to a multi-class hypothesis testing problem as follows: Consider we are given a set of $N$ score vectors $\{\boldsymbol{\theta}^1, \boldsymbol{\theta}^2, \ldots \boldsymbol{\theta}^N\} \subset \Theta_B(a, b)$ such that $\|\boldsymbol{\theta}^{k_1} - \boldsymbol{\theta}^{k_2}\| \ge \delta$, for any two score vectors $\boldsymbol{\theta}^{k_1}, \boldsymbol{\theta}^{k_2}$ such that $k_1, k_2 \in [N]$. Then given the set of pairwise preferences generated by an unknown sore vector $\boldsymbol{\theta} = \boldsymbol{\theta}^L$, where $L$ is a random index selected uniformly from the set $[N]$, the hypothesis testing task is to identify the index of the true score vector $L$.

Now given any algorithm that predicts a score vector $\hat{\boldsymbol{\theta}}$ based on the given set of pairwise preferences from the f-BTL model $\boldsymbol{\theta}^L$, sampled according to a $\mathcal{G}(n, p)$ Erdős-Rényi random graph with $p = \frac{\zeta}{n}$ for some $\zeta > 0$, such that $K$ independent noisy pairwise preferences are available for each sampled pair, one natural way to estimate $L$ is by $\hat{L} = \arg\min_{k \in [N]} \|\hat{\boldsymbol{\theta}} - \boldsymbol{\theta}^k\|$. Note that for $\hat{L}$ to be different that $L$, it has to be the case that $\|\hat{\boldsymbol{\theta}} - \boldsymbol{\theta}\| \ge \frac{\delta}{2}$. Thus one can write

$$\mathbf{E}[\|\hat{\boldsymbol{\theta}} - \boldsymbol{\theta}\|] \ge \frac{\delta}{2}\mathbf{P}(\hat{L} \ne L)$$

Further applying a similar information theoretic analysis as [16], one gets

$$\mathbf{E}[\|\hat{\boldsymbol{\theta}} - \boldsymbol{\theta}\|] \ge \frac{\delta}{2}\left[1 - \frac{\frac{K\zeta}{2N^2}\sum_{k_1 \in [N]}\sum_{k_2 \in [N]}\|e^{\boldsymbol{\theta}^{k_1}} - e^{\boldsymbol{\theta}^{k_2}}\|^2 + \log 2}{\log N}\right] \tag{24}$$

Thus the remaining task is to construct a set of $N$ score vectors $\{\boldsymbol{\theta}^1, \boldsymbol{\theta}^2, \ldots \boldsymbol{\theta}^N\} \subset \Theta_B(a,b)$ which are well separated, so to get suitable bounds on the terms $\|e^{\boldsymbol{\theta}^{k_1}} - e^{\boldsymbol{\theta}^{k_2}}\|^2, \ \forall k_1, k_2 \in [N]$ in (24). We use the following construction for the purpose:

**Constructing the set of score vectors.** For any $k \in [N]$, we construct the $k^{th}$ score vector $\boldsymbol{\theta}^k$ set of the set of $N$ random score vectors as follows:

- Draw $\alpha$ many random variables $X_1^k, X_2^k, \ldots X_\alpha^k \sim Unif\left[\left(\frac{1}{2} - \beta\delta\right), \left(\frac{1}{2} + \beta\delta\right)\right]$, where $\beta$ is a constant to be adjusted later.

- Set $\theta_i^k = a + (b-a)X_i^k, \ \forall i \in [\alpha], 0 < a < b < 1$.

- Consider the coefficient matrix $\mathbf{B} \in \mathbb{R}_+^{n \times \alpha}$ such that $\sum_{j=1}^\alpha B_{ij} = 1, \ \forall i \in [n]$.

- Set the remaining score vectors $\theta_i^k$ according to (2) for all $i \in [n] \setminus [\alpha]$.

We denote the restriction of the score vector $\boldsymbol{\theta}^k$ to the independent set $\mathcal{I}(G)$ by $\boldsymbol{\theta}_{[\alpha]}^k \in \mathbb{R}^\alpha$, where w.l.o.g. we assume $\mathcal{I}(G) = [\alpha]$ as before. Furthermore, from (2) for any two $k_1, k_2 \in [N]$, we have

$$\lambda_{\min}(\mathbf{B}^T\mathbf{B})\|\boldsymbol{\theta}_{[\alpha]}^{k_1} - \boldsymbol{\theta}_{[\alpha]}^{k_2}\|^2 \leq \|\boldsymbol{\theta}^{k_1} - \boldsymbol{\theta}^{k_2}\|^2 \leq \lambda_{\max}(\mathbf{B}^T\mathbf{B})\|\boldsymbol{\theta}_{[\alpha]}^{k_1} - \boldsymbol{\theta}_{[\alpha]}^{k_2}\|^2 \tag{25}$$

where $\lambda_{\min}(\mathbf{B}^T\mathbf{B})$ and $\lambda_{\max}(\mathbf{B}^T\mathbf{B})$ respectively denotes the minimum and maximum non-zero eigenvalues of the positive semi-definite matrix $\mathbf{B}^T\mathbf{B}$.

**Lemma C.1.** $\frac{1}{6}(b-a)^2\alpha\beta^2\delta^2 \leq \|\boldsymbol{\theta}_{[\alpha]}^{k_1} - \boldsymbol{\theta}_{[\alpha]}^{k_2}\|^2 \leq \frac{7}{6}(b-a)^2\alpha\beta^2\delta^2$, *for all* $k_1, k_2 \in [N] \times [N]$, *with probability at least* $(1 - N^2 e^{-\frac{\alpha}{32}})$.

*Proof.* Firstly we note that $\|\boldsymbol{\theta}_{[\alpha]}^{k_1} - \boldsymbol{\theta}_{[\alpha]}^{k_2}\|^2 = \sum_{i=1}^\alpha (\theta_i^{k_1} - \theta_i^{k_2})^2$ and for any $i \in [\alpha]$, $(\theta_i^{k_1} - \theta_i^{k_2})^2 = (b-a)^2(X_i^{k_1} - X_i^{k_2})^2$ and $\mathbf{E}[(X_i^{k_1} - X_i^{k_2})^2] = \frac{2}{3}\beta^2\delta^2$. Now applying Hoeffding's inequality we have that

$$\mathbf{P}\left(|\sum_{i=1}^\alpha (X_i^{k_1} - X_i^{k_2})^2 - \frac{2}{3}\alpha\beta^2\delta^2| \geq \frac{1}{2}\alpha\beta^2\delta^2\right) \leq 2e^{-\frac{\alpha}{32}},$$

for any fixed $k_1, k_2 \in [N] \times [N]$, and applying union bounding above holds true for all $\binom{N}{2}$ $(k_1, k_2)$ pairs with probability $N(N-1)e^{-\frac{\alpha}{32}} \leq N^2 e^{-\frac{\alpha}{32}}$. Now for any $N < e^{\frac{\alpha}{64}}$, we have $N^2 e^{-\frac{\alpha}{32}} < 1$ for all $\alpha > 0$, and hence with some non-zero probability of atleast $(1 - N^2 e^{-\frac{\alpha}{32}}) > 0$, we have

$$\frac{1}{6}\alpha\beta^2\delta^2 \leq \sum_{i=1}^\alpha (X_i^{k_1} - X_i^{k_2})^2 \leq \frac{7}{6}\alpha\beta^2\delta^2, \ \ \forall k_1, k_2 \in [N] \times [N].$$

Combining above we get,

$$\frac{1}{6}(b-a)^2\alpha\beta^2\delta^2 \leq \|\boldsymbol{\theta}_{[\alpha]}^{k_1} - \boldsymbol{\theta}_{[\alpha]}^{k_2}\|^2 \leq \frac{7}{6}(b-a)^2\alpha\beta^2\delta^2,$$

for all $k_1, k_2 \in [N] \times [N]$, with probability at least $(1 - N^2 e^{-\frac{\alpha}{32}})$. $\qquad\square$

For convenience let us fix $N = e^{\frac{\alpha}{128}}$. Thus using Lemma C.1 on (25), we get

$$\frac{\lambda_{\min}(\mathbf{B}^T\mathbf{B})}{6}(b-a)^2\alpha\beta^2\delta^2 \leq \|\boldsymbol{\theta}^{k_1} - \boldsymbol{\theta}^{k_2}\|^2 \leq \frac{7\lambda_{\max}(\mathbf{B}^T\mathbf{B})}{6}(b-a)^2\alpha\beta^2\delta^2,$$

with probability at least $(1 - e^{-\frac{\alpha}{64}})$. Now setting $\beta = \frac{\sqrt{6}}{(b-a)\sqrt{\alpha\lambda_{\min}(\mathbf{B}^T\mathbf{B})}}$ in above, we get

$$\delta^2 \leq \|\boldsymbol{\theta}^{k_1} - \boldsymbol{\theta}^{k_2}\|^2 \leq \frac{7\lambda_{\max}(\mathbf{B}^T\mathbf{B})}{\lambda_{\min}(\mathbf{B}^T\mathbf{B})}\delta^2, \text{ with probability at least } (1 - e^{-\frac{\alpha}{64}}) \tag{26}$$

**Lemma C.2.** *Given any two* $\boldsymbol{\theta}, \boldsymbol{\theta}' \in [a,b]^n$, *such that* $0 < a < b < 1$, *we have*

$$\|e^{\boldsymbol{\theta}} - e^{\boldsymbol{\theta}'}\|^2 \leq e^{2(b+1)}\|\boldsymbol{\theta} - \boldsymbol{\theta}'\|^2$$

*Proof.* The proof follows from the following straightforward deduction:

$$\|e^{\boldsymbol{\theta}} - e^{\boldsymbol{\theta}'}\|^2 = \sum_{i=1}^{n}(e^{\theta_i} - e^{\theta'_i})^2 = \sum_{i=1}^{n}(e^{\theta'_i})^2(e^{\theta_i - \theta'_i} - 1)^2$$

$$\leq \sum_{i=1}^{n} e^{2b}(e^{\theta_i - \theta'_i} - 1)^2 \leq e^{2b}\sum_{i=1}^{n}((\theta_i - \theta'_i)(e-1))^2$$

$$\leq e^{2(b+1)}\|\boldsymbol{\theta} - \boldsymbol{\theta}'\|^2,$$

where the second last inequality follows from the fact that $-1 < \theta_i - \theta'_i < 1$, for all $i \in [n]$. $\square$

We will now assume our constructed score vectors, $\boldsymbol{\theta}^k$, indeed satisfy $0 < a < \theta_i^k < b < 1, \forall i \in [n], \forall k \in [N]$. We will shortly show this is indeed true by our construction of $\boldsymbol{\theta}^k$. Then applying Lemma C.2 and subsequently C.1 to (24) we further get,

$$\mathbf{E}[\|\hat{\boldsymbol{\theta}} - \boldsymbol{\theta}\|] \geq \frac{\delta}{2}\left[1 - \frac{448e^{2(b+1)}K\zeta\Lambda\delta^2 + 128\log 2}{\alpha}\right], \tag{27}$$

where $\Lambda = \frac{\lambda_{\max}(\mathbf{B}^T\mathbf{B})}{\lambda_{\min}(\mathbf{B}^T\mathbf{B})}$, and $N = e^{\frac{\alpha}{128}}$.

Thus setting $\delta = \frac{\sqrt{\alpha}}{4\sqrt{448\zeta K\Lambda e^{2(b+1)}}}$, we have that

$$448e^{2(b+1)}K\zeta\Lambda\delta^2 + 128\log 2 \leq \frac{\alpha}{2}, \text{ for any } \alpha \geq 512\log 2,$$

using which in (27) further gives

$$\mathbf{E}[\|\hat{\boldsymbol{\theta}} - \boldsymbol{\theta}\|] \geq \frac{\delta}{4} = \frac{\sqrt{\alpha}}{16\sqrt{448\zeta K\Lambda e^{2(b+1)}}} = \frac{\sqrt{\alpha\lambda_{\min}(\mathbf{B}^T\mathbf{B})}}{16\sqrt{448\zeta K\lambda_{\max}(\mathbf{B}^T\mathbf{B})e^{2(b+1)}}}.$$

Finally, the only thing left to show is that indeed in the above construction of the score vectors $\boldsymbol{\theta}^k$ lies in the set $\Theta_B(a,b), \forall k \in [N]$. Note that if we can show $X_i^k \in [0,1], \forall i \in [\alpha]$, then that immediately implies $\theta_i^k \in [a,b], \forall i \in [n]$ by our construction of $\boldsymbol{\theta}^k$ and the assumption on the coefficient matrix $\mathbf{B} \in \mathbb{R}_+^{n\times\alpha}$ such that $\sum_{j=1}^{\alpha} B_{ij} = 1, \forall i \in [n]$.

Now we have $\left(\frac{1}{2} - \beta\delta\right) \leq X_i^k \leq \left(\frac{1}{2} + \beta\delta\right), \forall i \in [n]$ and $k \in [N]$. And with $\beta = \frac{\sqrt{6}}{(b-a)\sqrt{\alpha\lambda_{\min}(\mathbf{B}^T\mathbf{B})}}$ and $\delta = \frac{\sqrt{\alpha\lambda_{\min}(\mathbf{B}^T\mathbf{B})}}{4\sqrt{448\zeta K\lambda_{\max}(\mathbf{B}^T\mathbf{B})e^{2(b+1)}}}$, we have

$$\beta\delta = \frac{6}{4(b-a)\sqrt{448\zeta K\lambda_{\max}(\mathbf{B}^T\mathbf{B})e^{2(b+1)}}} < \frac{1}{2}.$$

Hence $0 \leq X_i^k \leq 1, \forall i \in [n]$ and indeed we have $\boldsymbol{\theta}^k \in \Theta_{\mathbf{B}}(a,b), \forall k \in [N]$. The desired lower bound now follows as:

$$\frac{\mathbf{E}[\|\hat{\boldsymbol{\theta}} - \boldsymbol{\theta}\|]}{\|\boldsymbol{\theta}\|} \geq \frac{\sqrt{\lambda_{\min}(\mathbf{B}^T\mathbf{B})}}{16b\lambda_{\max}(\mathbf{B}^T\mathbf{B})\sqrt{448\zeta K e^{2(b+1)}}},$$

since $\|\boldsymbol{\theta}\| \leq \sqrt{\lambda_{\max}(\mathbf{B}^T\mathbf{B})}\|\boldsymbol{\theta}_{[\alpha]}\| \leq b\sqrt{\lambda_{\max}(\mathbf{B}^T\mathbf{B})\alpha}$.

$\square$