# OpenReview forum: "A Graph Theoretic Approach for Preference Learning with Feature Information"
_auai.org/UAI/2024/Conference — UAI 2024 oral_

### Official Review · Reviewer_UT5e · 2024-03-19

**Q2-1 Originality-Novelty:** 3
**Q2-2 Correctness-Technical Quality:** 3
**Q2-5 Clarity Of Writing:** 3

**Q10 Ethical Concerns:**

I do not anticipate any ethical concerns for this paper.

**Q1 Summary And Contributions:**

The paper studies the problem of learning a global ranking of n items based on a random sample of pairwise comparisons. In general, this takes Omega(n log n) comparisons if you get to choose the pairs, or Omega(n^2) if you don’t. The present paper asks whether these bounds can be improved if the comparisons are based on each item’s feature vector, and the feature vectors have a basis of dimension $\alpha$ (where one imagines $\alpha$ < n). The paper assumes that the basis vectors are themselves feature vectors for $\alpha$ of the n items. The model also allows noisy comparisons in the style of the Bradley-Terry-Luce model.

The authors provide an algorithm with sample complexity O($\alpha$ log $\alpha$), along with a matching lower bound. The authors experimentally validate their results on both real and synthetic data. The paper also provides some analysis of the case with noiseless comparisons. The theoretical tools are primarily based on the underlying “relation graph”, where the low-dimensional basis is represented as an independent set.

**Q2-3 Extent To Which Claims Are Supported By Evidence:**

3: Good: the main claims are supported by convincing evidence (in the form of adequate experimental evaluation, proofs, (pseudo-)code, references, assumptions).

**Q2-4 Reproducibility:**

3: Good: key resources (e.g. proofs, code, data) are available and key details (e.g. proofs, experimental setup) are sufficiently well-described for competent researchers to confidently reproduce the main results.

**Q3 Main Strengths:**

*Motivation and assumptions*. Ranking is certainly an important problem, and the feature vector model seems realistic for most practical applications. The low-dimensional basis assumption also seems fine, although it seems like the algorithm needs to know which $\alpha$ items make up the basis vectors, which is not ideal. (See e.g. the use of $\alpha$ in Algorithm 1.)

This concern is somewhat alleviated by the experiments on real data, where the algorithm does not know upfront which $\alpha$ items to use for a basis, and instead must find a basis based on observations. The algorithm achieves good performance despite this. (For these experiments, $\alpha$ is assumed to be the dimension of the feature vectors.)

*Novelty*. The paper makes a non-trivial advance over prior work which obtained sample complexity O($\alpha$^2 log n) [16], although [16] does not require the basis vectors to be among the items’ feature vectors. It is laudable that the experiments include the algorithm from [16] and show that the present algorithm outperforms it empirically.

*Techniques*. I found the graph-theoretic techniques to be interesting, although not groundbreaking.

*Presentation*. The paper is fairly well-written overall (see below for some comments).

**Q4 Main Weakness:**

1. Even with the experiments on real data, the fact that the theoretical results require the algorithm to know an $\alpha$-item basis a priori is a weakness. I am curious if the authors think it is possible to generalize their theoretical results to eliminate that assumption, e.g. by learning a basis based on observations (as they do in the experiments on real data).

2. I think the comparison with related work could be improved. First of all, this section does not appear until the appendix, which is unfortunate, as it is important for evaluating the novelty of the present paper. I also think a more thorough comparison with [16] would be valuable. The authors state “The most related work is [16], however they assume the features to lie in some low-dimensional space and use a matrix completion based approach to predict the ranking.” It seems to me that in effect this paper also assumes the features lie in a low-dimensional (sub)space, and thus the results from [16] should apply to the present paper’s model. However, the bound in [16] of O($\alpha$^2 log n) is worse than the present paper’s bound of O($\alpha$ log $\alpha$), which seems like it should be a primary selling point.

3. I don’t feel that the synthetic data experiments contribute much to the paper. It seems like the data is specifically generated to match the assumptions of the theoretical model, in which case of course the algorithm works. I don’t think these experiments detract from the paper, but I’m not sure they’re worth including in the main body. In particular, I would consider replacing Section 6 with the real data experiments (which are currently in the appendix).

4. Some of the technical writing could be improved. The paper is quite technically dense, and the authors don’t provide much exposition or intuition for their proofs. Also, the “Remarks” sections at the end of Sections 4 and 5 should be full theorems. If you state that your primary result is a O($\alpha$ log $\alpha$) bound, the reader expects to see a theorem stating that.

**Q5 Detailed Comments To The Authors:**

*Main comments*

1. My main question is whether it’s possible to do away with the whole “$\alpha$ independent items framing” and just say that the feature vectors have an $\alpha$-dimensional basis, or even just say that the feature vectors are in R^$\alpha$, which I think many readers would find more intuitive. Is there a reason this other framing is necessary? It is important that the $\alpha$ basis vectors actually belong to some $\alpha$ items?

2. As mentioned above, does the algorithm really need to know the basis upfront?

*Minor comments*
1. Abstract: “makes use of tools from classical graph matching theory, shedding light on the true complexity of the problem – this was not possible before with existing matrix completion based tools”. This is pretty vague and I’m not sure it conveys much to the reader.
2. Pg 1: “Secondly, they fail to handle the case when new items get added". This was confusing to me because it sounds like your model is going to allow for items to be dynamically added over time, which it doesn’t.
3. The description of Section 3 in the left column of page 2 seems off. Shouldn’t it mention that Section 3 is specifically about noiseless preferences? I also wonder if Section 3 should go at the end, it seems somewhat less important compared to the main O($\alpha$ log $\alpha$) result.
4. In “Notations” shouldn’t A be bolded?
5. “we also assume B is such that any $\alpha$ × $\alpha$ submatrix of B is of rank $\alpha$” can be simplified to “we aso assume that any $\alpha$ × $\alpha$ submatrix of B is of rank $\alpha$”. Same with “G([n], E) is such that…”
6. In Eq 1, do you need the condition that (i,j) in E? Can’t you just set B_ji = 0 for those pairs? Then you can just write sum_{j in I(G)} B_ji u_j.
7. “Our work has no societal impact.” One would hope that isn’t true! Consider something like “we do not anticipate any problematic societal repercussions from our work.”
8. Starting sentences with a numerical reference is awkward in my opinion. For example, “[1] study X. [2] study Y.”
9. It seems like Section 3 assumes that B, U, and P are all known, but you only state that “...we have access to the exact value for P_ij”. I would suggest clarifying what the assumption here is.
10. Theorem 3.1: “a unique solutions” → “ a unique solution”
11. Figure 3 has some formatting issues: (A) The rightmost label on the x-axis is cut off on some of the plots (e.g. it says 50 but should say 500), (B) the legend is cut off on some of the plots, and (C) only some of the plots have error bars
12. I would suggest briefly noting how error bars were computed (is it just bootstrapping?)

**Q9 Complying With Reviewing Instructions:**

Yes

---

> ### Author Rebuttal · Authors · 2024-04-06
>
> Thank you for your careful reading and insightful feedback.
>
> >> Q1: Is it possible to do away with the whole “\alpha independent items framing” and just say that the feature vectors have an $\alpha$-dimensional basis, or even just say that the feature vectors are in $\mathbb R^\alpha$? Is there a reason this other framing is necessary? It is important that the basis vectors actually belong to some items?
>
> -- Thanks for your thoughts and the very interesting question! You are absolutely right that the main structural assumption we exploited in our main results (Sec 4, 5, 6) is the low $\alpha$-dimensional embedding, towards yielding a $O(\alpha \log \alpha)$ sample complexity. The graph theoretic interpretation of Eq1 and $\alpha$ being the independence number of the underlying relation graph can be generalized as described below without affecting our algorithms and results in Sec 4,5.
>
> Indeed, for a more general overview of our problem setup, we could have simply assumed $\mathcal I(G)$ to be an index set of basis items, where the set {$ \{u_i \in \mathbb{R}^{\alpha} \mid {i \in \mathcal{I} (G) }\} $} represents a basis of the true feature set{$U$}. Further, to mimic Eq(1), now we assume a corresponding coefficient matrix $\tilde B$ s.t. $U = \tilde B U_{\alpha}$, where $U_\alpha$ represents the "basis matrix" with vectors in {$u_i \mid i \in \mathcal I(G)$} stacked in the columns of $U_\alpha$.
>
> In fact, note we do not need the knowledge of $U_\alpha$ apriori: As given the true feature matrix $U$, we can derive one basis (by Gauss elimination or even Gram-Schmidt) that spans the feature space set{$U$}. As you correctly pointed out, this is precisely what we adapted for our real-data experiments as well.
>
> For our main upper and lower bounds results in Sec 4 and 5, note it is not even necessary for a set\{$U_\alpha$\} $\subset$ set\{$U$\} or that the "basis vectors should belong to an item". Note $\{u_i\}$ s.t. ${i \in \mathcal I(G)}$ could very well be the features of some hypothetical items--the only important factor towards the improved $O(\alpha \log \alpha)$ sample complexity being the existence of the coefficient matrix B (i.e. the low $\alpha$-dimensional embedding of the feature vectors set\{$U$\}.
>
> We will add a remark at the end of Sec 4 to clarify this more general view of our problem setting and how our algorithm (Alg 1) exploits the lower dimensional feature embedding to yield the improved sample complexity rate.
>
> ---
>
> -- Lastly to answer "Is there a reason this other framing is necessary?": The current framing of our problem setup, the maximal independent set-based formulation gives a nice graph theoretic flavor to our problem setting yielding the results in Sec 3 (Thm 3.2, 3.3, etc), which highly relies on Hall's Marriage theorem (and consequently Thm 3.1). Thm 3.2 shows how we can recover the true $\theta$ with high probability with a few (noiseless) pairwise observations, and specific results for some special family of graphs (Thm 3.3). This section requires the maximal independent set-based formulation of Eq (1), and hence we framed our problem setting based on the graph-theoretic interpretations and later generalized it in Sec 4 and 5.
>
> >> Q2: Related Work and comparison with [16].
>
> We will certainly put the related work in the main draft (given are allowed to use 2 extra pages in the final version). You are also absolutely right about comparing our work. "$d$" in [16] is indeed equivalent to our "$\alpha$" (embedding dimension), as we already clarified our problem setups in Q1 above. We also compared our performances in Table 1 (Appendix), which we will add to the main draft. Indeed, as you pointed, we improve $O\big(d \frac{\log n}{\log \alpha} \big)$ factor compared to [16].
>
> >> Q3: Consider replacing Section 6 with the real data experiments.
>
> Thanks for the suggestions. Certainly, we agree and we will add the real-data experiments to the main draft (in Sec 6), since we will also be allowed to include have 2 extra pages in the final version.
>
> >> Q4: Improving technical writing:
>
> Many thanks for the suggestions. We will make the proof sketches more detailed and provide intuitions. Further, it's a great suggestion to make the two remarks at the end of Sec 4 and 5 as full theorems, indeed that will precisely reflect our $O(\alpha \log \alpha)$ sample complexity guarantee of Alg1 (fBTL-LS). We will make these changes in the final version.
>
> >> Q5: Do we need to know the basis upfront?
>
> As we also explained in Q1 above, we don't need to know the basis upfront. Given the feature matrix $U$, we could derive a basis of the feature vectors and hence the coefficient matrix $B$. Please see our response in Q1 for a detailed discussion.
>
> We will be happy to discuss any additional comments you may have. Please let us know.
>
> Thanks
> Authors
>
> ---
> ---
> Also thanks for the minor comments. We will incorporate all the suggestions and also answer the minor comment qs in the follow-up below. Thank you.

---

### Official Review · Reviewer_odST · 2024-03-22

**Q2-1 Originality-Novelty:** 3
**Q2-2 Correctness-Technical Quality:** 3
**Q2-5 Clarity Of Writing:** 4

**Q1 Summary And Contributions:**

In this paper, the authors study the problem of ranking items given a sample of pairwise preferences. In particular, they study the setting where more information is known about the internal structure of the items (i.e., features), and the relation between items in terms of their features. They introduce a new probabilistic model to capture this setting, and an algorithm that produces a ranking of high quality with much smaller than the 'unstructured' case. In addition to these theoretical results, they experimentally test their approach on various data sets.

**Q2-3 Extent To Which Claims Are Supported By Evidence:**

3: Good: the main claims are supported by convincing evidence (in the form of adequate experimental evaluation, proofs, (pseudo-)code, references, assumptions).

**Q2-4 Reproducibility:**

4: Excellent: key resources (e.g. proofs, code, data) are available and key details (e.g. proof sketches, experimental setup) are comprehensively described for competent researchers to confidently and easily reproduce the main results.

**Q3 Main Strengths:**

- The authors introduce a new model for a realistic and relevant setting, and provide an algorithm that performs well in this case
- They give a solid theoretical analysis of the new setting
- They show that the theoretical performance is matched in practice by means of experiments

**Q4 Main Weakness:**

(none)

**Q5 Detailed Comments To The Authors:**

I find the paper to be a solid contribution. The paper is well written: it is clear what problem the authors set out to solve, and how they solve it. The model seems natural and seems to match reality nicely. The model is not overly complicated, but adds exactly the right elements to allow for better modelling/performance. The theoretical results are clear, and the proofs are non-trivial. (I did not manage to check all proofs in full detail, but I found no errors.) The experimental results corroborate the theoretical results, and the experiments are described clearly.

**Q9 Complying With Reviewing Instructions:**

Yes

---

> ### Author Rebuttal · Authors · 2024-04-06
>
> Thanks a lot for your careful reading, positive comments, and appreciating our work. We will be happy to discuss any additional comments that you may have. Please let us know.
>
> Thanks
> Authors

---

### Official Review · Reviewer_Qh6u · 2024-03-24

**Q2-1 Originality-Novelty:** 3
**Q2-2 Correctness-Technical Quality:** 3
**Q2-5 Clarity Of Writing:** 2

**Q1 Summary And Contributions:**

The goal of object ranking with pairwise preferences is to rank n items based on m pairwise preferences between them, which are tested K times. The Bradley-Terry model is usually used, where pairwise probabilities are defined by a parameter vector $\theta$ that assigns a score to each item. A new variant of this model is proposed in this paper, which considers the feature vector $u_i$ of each item $i$. The key concept of this study lies in the concept of "relation graph" $G$ of order $n$, which captures some regularities in the feature matrix $U$. The study assumes that $U$ can be inferred from an independent set $I(G)$ of $G$ and a coefficient matrix $B$. The authors show that the sample complexity of the preference learning task essentially depends on the size of the independent set. A matching lower bound is also provided. Numerical simulations on synthetic benchmarks support the theoretical results.

**Q2-3 Extent To Which Claims Are Supported By Evidence:**

2: Fair: the main claims are somewhat supported by evidence (but the experimental evaluation may be weak, or does not match entirely with the claims, important baselines may be missing, proofs contain important ideas but lack rigor, algorithmic details are only discussed superficially, references are imprecise, assumptions are not sufficiently motivated or explicated, etc.).

**Q2-4 Reproducibility:**

3: Good: key resources (e.g. proofs, code, data) are available and key details (e.g. proofs, experimental setup) are sufficiently well-described for competent researchers to confidently reproduce the main results.

**Q3 Main Strengths:**

The idea of incorporating features into the Bradley-Terry model is natural, and very interesting. From this viewpoint, the paper is well-motivated and I agree with the authors that this feature-based Bradley-Terry model has many potential applications. In addition, the idea of reducing the sample complexity of the learning problem using an independance graph from which correlations are inferred is also quite natural. The study is comprehensive, covering both known and unknown preferences, a matching lower bound, and numerical simulations.

**Q4 Main Weakness:**

The readability of the paper might be improved. Notably, the definition of the relation graph $G$, and the two main assumptions (independent set, and matrix rank) could be rephrased in a more intuitive way. Another potential weakness is the complexity of the following problem: given a matrix $U$ of dimension $n \times n$, does $U$ admit a relation graph $G$ satisfying the assumptions in Section 2.1? From this viewpoint, it would be helpful to include a discussion about the complexity of this problem, as well as the runtime complexity of the main algorithm (fBTL-LS).

**Q5 Detailed Comments To The Authors:**

I have taken some time to understand how to derive $U$ from $I(G)$ and $B$ using Equation (1). It is important to note that if $I(G)$ is not maximal, then it is possible that the intersection of $I(G)$ and the neighborhood of $i$, extended to $i$, is empty. This means that the matrix $U$ is not well-founded in that case. However, if $I(G)$ is a maximal independent set, then the definition looks correct. In that case, the value $\alpha$ capturing the sample complexity of the learning problem is indeed the independent number of $G$. Unfortunately, the problem of finding a maximal independent set is NP-hard and hard to approximate to within a constant. Therefore, if $I(G)$ must be maximal, then testing whether a given feature matrix $U$ satisfies the relation-graph conditions is also intractable. This can significantly reduce the practical usefulness of the feature-based Bradley-Terry model. I would appreciate any comments about this issue.

**Q9 Complying With Reviewing Instructions:**

Yes

---

> ### Author Rebuttal · Authors · 2024-04-06
>
> Thank you for appreciating our problem-setting and the insightful comments.
>
> >>Q1. In Eqn (1), is $\mathcal I(G)$ is maximal IS?
>
> Yes, we meant $\mathcal I(G)$ is the maximal IS, as otherwise $\mathcal I(G) \cap \bar N(i)$ could be a null set, giving rise to vacuous settings.
>
> >> Q2. Does $U$ admit the relation in Eq (2.1)? Complexity of the problem, and runtime complexity of the main algorithm Alg 1 (fBTL-LS).
>
> Alg1 (FBTL-LS) assumes $U$ is known, as generally, it is a standard and realistic assumption to consider the feature space is learner's knowledge [3,11,15,16], [[Bandit Algorithms]](https://banditalgs.com/). E.g. in the mobile phone example in Sec 1, one may assume that the phone manufacturer will know the features (specifications) of their phones but the utility mapping function $\mathbf w$ and hence $\theta_i$s are unknown.
>
> But even if $U$ and $G$ is given, it is hard to find the exact $B$ in Eq1 unless the corresponding $\mathcal I(\alpha)$ is known. As correctly pointed out, given any graph $G$, finding a maximal independent set (MIS) of G (or even a constant fraction approximation) is NP-hard (except some [special family](https://graphclasses.org/) of graphs). But more importantly, to find the exact $B$ in Eq(1), we actually need to know the specific $\mathcal I(G)$ (since MIS is not necessarily unique), which is impossible unless $\mathcal I(G)$ is provided to the learner. We could assume $\mathcal I(G)$ is also known but it's not necessary. The fact is:
>
> **To run Alg1, we do not need to know the exact $\mathcal I(G)$ and B of Eq(1), but it is sufficient to know any basis of the feature set$\{U\}$ and its corresponding coefficient matrix B.** Thus we do not need the knowledge of any MIS given $U$. Of course, the given $U$ admits the relation in Eq(1).
>
> Indeed, for a more general overview of our problem setup (Eq(1)) is where we could simply assume that $\mathcal I(G)$ is any an index set of basis items, where the set {$u_i \in \mathbb R^\alpha \mid i \in \mathcal I(G)$} represents a basis of the true feature set{$U$}. Further to mimic the structure of Eq(1), now we can assume that there exists a corresponding coefficient matrix $\tilde B$ such that $U = \tilde B U_{\alpha}$, where $U_\alpha$ represents the "basis matrix" with the set of vectors in {$u_i \mid i \in \mathcal I(G)$} being stacked in the columns of $U_\alpha$.
>
> In fact, note that we do not need the knowledge of $U_\alpha$ apriori. This is because given the true feature matrix $U$, we can derive one such basis (by Gauss elimination or even Gram-Schmidt) that spans the feature space set{$U$}. Further, the size of such a basis has to be $\alpha$ since $U$ satisfies Eq(1).
>
> Now given $U$, Alg 1 could simply find such a basis $U_{\alpha}$ and its corresponding coefficient matrix $\tilde B$ and run the remaining steps of Alg1 as is. Since $U_\alpha$ and $\tilde B$ satisfy Eq(1), the analysis of Alg1 will follow as is.
>
> To summarize, we only assume that the underlying feature representation of the [n] items follows a low $\alpha$-dimensional feature embedding given by Eq(1), and by virtue of that Alg1 can derive a basis  set$\{U_\alpha\}$ (such that |\{$U_\alpha$\}| = \alpha) and the corresponding coefficient matrix $\tilde B$ satisfying $U = \tilde B U_{\alpha}$. Alg1 can simply work with $\tilde B$ henceforth, yielding the exact same analysis and performance guarantee of Thm 4.1.
>
> This general view of Eq(1) could be more convincing for a practical viewpoint which simply implies that the feature set of the n items lies in a low $\alpha$-dimensional embedding. We thank you for the question, and we will certainly clarify the above details and the general view of our problem setup at the beginning of Sec 4 and Alg1.
>
> (Please also see our response for Q1 of Reviewer QT5e for related discussions).
>
> **Re. Runtime Complexity of Alg1 (fBTL-LS):** We already discussed the knowledge and derivation of $U$ and $B$ above. Given $U$ and $B$ the rest of the steps of Alg1 are computationally efficient as they are simply based on statistical estimation of $P_{ij}$, solving linear equations and convex programming, leading to only poly(m) complexity of Alg1 (fBTL-LS).
>
> -- Finally, we also thank you for the suggestions on improving the readability of the paper, we will expand on the definition of the relation graph, and the two main assumptions (independent set, and matrix rank) with more examples. Since we will be allowed to use 2 additional pages in the final version, we will add all the above details in the final main draft.
>
> We will be happy to discuss any additional comments that you may have. Please let us know.
>
> Thanks
> Authors

---

### Official Review · Reviewer_rgcR · 2024-03-27

**Q2-1 Originality-Novelty:** 3
**Q2-2 Correctness-Technical Quality:** 4
**Q2-5 Clarity Of Writing:** 3

**Q1 Summary And Contributions:**

The paper addresses the problem of ranking items based on pairwise preferences when item features are available to improve the efficiency of the ranking problem. ​ The key contributions include the introduction of the f-BTL model, which uses features to estimate item scores, and the development of the fBTL-LS algorithm, a least squares implementation of f-BTL. ​ The paper provides substantive theoretical analysis of the algorithm's sample complexity and recovery guarantees. A small set of experiments are provided which show strong performance of the proposed approach.

**Q2-3 Extent To Which Claims Are Supported By Evidence:**

4: Excellent: all claims are supported by very convincing evidence (in the form of comprehensive experimental evaluation, rigorous mathematical proofs, detailed (pseudo-)code, precise references, well-motivated and realistic assumptions) and the authors deliver what they promise.

**Q2-4 Reproducibility:**

3: Good: key resources (e.g. proofs, code, data) are available and key details (e.g. proofs, experimental setup) are sufficiently well-described for competent researchers to confidently reproduce the main results.

**Q3 Main Strengths:**

1. Paper is well written and organized. The authors do a nice job of clearly laying out the problem setting, explaining the proposed solution and its properties.
2. The authors provide extensive theoretical analysis of the proposed methods, matching a lower bound derived in the paper.
3. Strong empirical performance.
4. The problem is well motivated by application.

**Q4 Main Weakness:**

Overall, I think this a very nice, strong paper. The empirical evaluation is a little bit weak. It would have been nice if the algorithms could be evaluated on real world data, or at least semi-synthetic data.

**Q5 Detailed Comments To The Authors:**

As I mention above, I think this is an interesting contribution to the literature. The sample complexity results are quite interesting, and the proposed algorithms are both simple and performant. As I mentioned above it would be nice if the authors could provide more extensive real-world (or real world approximation) experiments.

**Q9 Complying With Reviewing Instructions:**

Yes

---

> ### Author Rebuttal · Authors · 2024-04-06
>
> Thank you for appreciating our work and your feedback.
>
> >>Q1. Real-world or semi-synthetic data:
>
> Thanks for the suggestion, however, please note that we indeed reported some experiments on two real datasets in Appendix E.1 (as mentioned in the last paragraph of Sec 6), where also our algorithm (fBTL-LS) could be seen to outperform the remaining baselines.
>
> We will be happy to include more real-data experiments if the reviewer has any additional datasets for suggestion. Additionally, we will also move the real data experiments (in Appendix E.1) to the main paper, as in the final version UAI allows "the main part up to 10 pages (excluding references and appendices)".
>
> We will be happy to discuss any additional comments that you may have. Please let us know.
>
> Thanks
> Authors

---

### Meta-Review · Area_Chair_ongG · 2024-04-19

The paper discusses the classical problem of item ranking given samples of pairwise preferences with a twist: the items have features, and there is side-information about the feature relationships between items, ie, they lie in a low-dimensional subspace of dimension $\alpha$. They show how to exploit this side information so as to bring down sample complexity from $O(n \log n)$ to $\theta(\alpha \log \alpha)$.

The reviewers all agree that this is a valuable and interesting contribution. The weaknesses identified by reviewers mainly concern questions of writing clarity and additional/more thorough evaluations that could have been done. All recommend acceptance, and I agree with this assessment.


PS: One more reference that might be related is Kadioglu et al, "Sample complexity of rank regression using pairwise comparisons", Pattern Recognition Oct 2022.